# Comparative optimization of polysaccharide-based nanoformulations for cardiac RNAi therapy

Han Gao[1,2,11], Sen Li[3,11], Zhengyi Lan[4], Da Pan [5], Gonna Somu Naidu[6,7,8,9], Dan Peer [6,7,8,9], Chenyi Ye[10], Hangrong Chen [4], Ming Ma [4] ✉, Zehua Liu [1,2] ✉ & Hélder A. Santos [1,2] ✉

Ionotropic gelation is widely used to fabricate targeting nanoparticles (NPs) with polysaccharides, leveraging their recognition by specific lectins. Despite the fabrication scheme simply involves self-assembly of differently charged components in a straightforward manner, the identification of a potent combinatory formulation is usually limited by structural diversity in compound collections and trivial screen process, imposing crucial challenges for efficient formulation design and optimization. Herein, we report a diversity-oriented combinatory formulation screen scheme to identify potent gene delivery cargo in the context of precision cardiac therapy. Distinct categories of cationic compounds are tested to construct RNA delivery system with an ionic polysaccharide framework, utilizing a high-throughput microfluidics work-station coupled with streamlined NPs characterization system in an automatic, step-wise manner. Sequential computational aided interpretation provides insights in formulation optimization in a broader scenario, highlighting the usefulness of compound library diversity. As a result, the out-of-bag NPs, termed as GluCARDIA NPs, are utilized for loading therapeutic RNA to ameliorate cardiac reperfusion damages and promote the long-term prognosis. Overall, this work presents a generalizable formulation design strategy for polysaccharides, offering design principles for combinatory formulation screen and insights for efficient formulation identification and optimization.

Pioneering non-viral nanoparticles (NPs) for delivering genetic cargos, such as small interfering RNAs (siRNA), opens up opportunities for novel therapeutic paradigm[1]. In the relevance of vehicle design, along with the requirements like oligonucleotides encapsulation/protection, biocompatibility and efficient intracellular delivery, precise targeting is gaining particular focus due to the growing interests in delivering siRNAs beyond the liver[2].

Different strategies, including tuning the internal/external charge with different lipids to achieve specific organ tropism[3], targeting ligand modification[4] or manipulating protein corona on the surface[5], have been adopted to mediate target cell entry in desired site. Of note,

as an alternative method, previous studies proposed a simplified NPs design principle by tailoring the choice and chemical structure of polysaccharides as building blocks to facilitate the targeted delivery[6–9]. This polysaccharide framework capitalizes the exclusive recognition of certain glycans by specific lectins, which may be abnormally expressed in lesioned sites, enabling a targeted delivery without subsequent modifications.

One common method to construct polysaccharide NPs in oligo-nucleotide delivery is through ionotropic gelation, showing advantages in simple and green synthesis scheme[10]. Leveraging the interactions between polysaccharide and oppositely charged

counterionic species, various polysaccharides, such as chitosan and alginate, have been extensively studied for fabricating siRNA carriers[10,11]. However, to date, much of these works have centered on formulation optimization in specific ionotropic pairings, which lacks comparative analyses between different ionotropic pairings for more efficient formulation optimization. In contrast, a "diversity-oriented" combinatorial optimization approach, which involves screening a distinct collection of molecules characterized by a wide spectrum of chemical structures, may expand the range of potential ionotropic interactions, and facilitate a parallel comparison between different pairings. By diversifying the investigated chemical interactions, this methodology enhances the probability of uncovering unique and optimum candidates for siRNA carrier development. Furthermore, a rationale gene delivery system design and screen necessitate the evaluation of various factors, including particle size, encapsulation efficiency, biocompatibility, and gene transfection efficiency. However, a systematic and step-wise evaluation of influencing factors at each step remains undefined, and more importantly, the physiochemical mechanism beneath the screening process remains less understood. Therefore, further efforts in identifying the informative physiochemical descriptors of chemical matters for formulation optimization is highly desired, and the developed method is preferably in an interpretable manner to a credential scientific understanding, which holds particular value in medical applications[12].

In this study, a step-wise screening system was designed to comprehensively identify the optimal polysaccharide-based nanoformulation for targeted gene delivery (Fig. 1). For comparative optimization of diverse ionotropic pairings in constructing NPs, we selected representative materials as cationic parts, including lipids, polymers, small molecules and cell-penetrating peptides, whereas the endosomolytic phosphorylated β-glucan (EEPG) was fixed as anionic framework (Supplementary Fig. 1)[6]. An automated microfluidics platform was applied to facilitate the screening process, followed by systematically evaluating the NPs formation capability, biocompatibility, siRNA encapsulation efficiency and gene knockdown efficiency of the nanoformulations. Upon this, we adopted computational analysis to sequentially interpret each step during the screening process, which can provide potential understanding for NPs formation process and identify the key physiochemical parameters in decision-making. Consequently, we ended by identifying an optimal candidate for targeted siRNA delivery, termed as GluCARDIA. Taking advantage of the inherent affinity of EEPG towards Dectin-1[+], which is over-expressed in injured myocardium, the proposed GluCARDIA showed superior cardiac accumulation with efficient gene silencing efficacy after myocardial ischemic/reperfusion (IR) injury. As a proof-of-concept, the tailored RNAi nanosystem was tested in a clinical-relevant murine myocardial IR model. Overall, this study presents a step-wise, diversity-oriented methodology for screening polysaccharide-based nanoformulations, advancing the era of RNAi therapeutics in ischemic heart diseases.

## Results

### A robotic assisted microfluidics platform for nanoparticles formation screening

An ideal RNAi delivery system was customized to screen and determine the feasibility and physical properties over a range of dynamic parameters. Followed with our design criteria, a rapid, one-step synthetic method was developed through the high-throughput microfluidics workstation coupled with streamlined dynamic light scattering (DLS) analysis system (Fig. 2a, Supplementary Fig. 2; Supplementary Video S1). Such integrated NPs' production-characterization system automates the screen process while maintain the reproducibility, which is ideal for in vitro screening of NP-formulations. Briefly, EEPG was immobilized as ionic part, which was further co-assembled with oppositely charged molecules, including polymers, lipids, small

molecular drugs and cell-penetrating peptides (Fig. 2b). The fabrication of EEPG-based NPs was achieved by microfluidics assisted nanoprecipitation method, where EEPG was fixed as outer phase and different cationic molecules were adopted as inner phase, and the NPs formation was triggered by passive microfluidic mixing in a co-flow microfluidic devide[13,14]. By changing flow conditions of injection system, different weight ratios or nitrogen: phosphate (N/P) ratios between cationic materials and EEPG were evaluated and simultaneously analyzed by the online size characterization system, which integrates an automatic dynamic light scattering (DLS) instrument. The detailed information in collection system (solvents, flow conditions, inner phase versus outer phase) are listed in Supplementary Table 1. As such, multiple formulations were tested, and the average derived counting rate over 6000 kcps was indicative to the formation of NPs. The overall screening results are summarized in Fig. 2c. The formation of stable NPs is shown as blue dot, whereas grey dot indicates the weak interaction of the two compartments without pervasive formation of NPs.

In the screening pool, we set 7 different weight ratios (w/w) between each category of cationic biomaterials and EEPG for our initial screens. From the standpoint of N/P ratios, all the tested w/w ranges covered the N/P ratio at 1/1, where the intermolecular electrostatic interaction between protonated amino groups and corresponded phosphorylated groups was expected to be maximally evident[15]. For the category 1 (C1), cationic polymer-based biomaterials, due to their significant variation in molecular weight and amino substitution rate among different polymers[16], we chose to adopt the corresponding N/P ratios to better interpret the NPs formation in an ionotropic gelation scenario. The detailed information regarding weight ratios and corresponded N/P ratios in each subpanel is shown in Supplementary Table 2–5. The hydrodynamic diameter and zeta potential of representative nanoformations is shown in Fig. 2d, e, and promising pairings from each category are imaged by transmission electron microscopy (TEM) (Supplementary Fig. 3).

For C1, we noticed that under a same N/P ratio, amine-modified dextran failed to form NPs with EEPG at all tested sets, whereas other cationic polymers, including chitosan and PEI, can feasibly form NPs under the N/P ratio near 1/1, which is consistent with previous studies[17]. In C2, which represents the small molecular drugs, despite their highly enriched amine groups in molecular structure, none of them can form into stable NPs around N/P 1/1, where spermine can only form into NPs at high weight ratio (8/1, corresponding to N/P = 30/1). For cationic cell-penetrating peptides (C3), we observed pervasive NPs formation in all tested molecules at weight ratio 1/1 and 1/2 (which covered the N/P range 1/1), where K9 and KALA can also form into NPs at weight ratio 8/1. In C4, which represents cationic lipids, despite Lipid 8 and 10 cannot form into NPs with EEPG at the N/P around 1/1, they can form into stable NPs with EEPG with extra high amount of EEPG, where the N/P is lower than 0.04/1, whereas Lipid 14 cannot form NPs with EEPG in all tested formulations (Supplementary Table 6).

Next, a subset of 16 nanosystems with the most optimal physical characters (smallest size with the PDI < 0.2) were adopted for further screen and evaluations (Fig. 2d, e).

### siRNA encapsulation efficiency of nanoformulations

Efficient siRNA encapsulation and protection is another critical aspect for formulation optimization, therefore we further assessed the siRNA encapsulation efficiency using afore-defined nanosystems with different cationic species, and Ribogreen assay was initially adopted for quantifying the RNA encapsulation[18]. Previous studies emphasize the efficient encapsulation of siRNA majorly relies on the electrostatic interaction between cationic biomaterials and negatively charged phosphate backbone of nucleic acid, as indicated by the (N/P) ratio[11]. Therefore, we evaluated the siRNA encapsulation efficiency of afore-screened formulations at fixed N/P ratios of 15/1 or 20/1 respectively

High-throughput microfluidic platform for formulation screening

**Fig. 1 | Schematic illustration of workflow for the current study.** The current research arose with the construction of cationic compounds collections with distinct categories, and a previously synthesized multi-functional polysaccharide was fixed as ionic part. The screen process was aided by a high-throughput microfluidics workstation coupled with streamlined NPs characterization system, and the step-wise screen was interpreted by a computational aided analysis for potentially understanding the decision-making mechanism. In the end, the out-of-bag RNAi nanomedicine was tested in a murine myocardial reperfusion model to evaluate its corresponding therapeutic potency.

(Figs. 3a, b). As expected, under the same N/P ratio, the selection of cationic compounds significantly affected the encapsulation efficiency of siRNA. Meanwhile, for most of the nanosystems, the encapsulation efficiency (EE%) was positively correlated with N/P ratio, with the exception of Transportan-EEPG NPs, where we noticed a significant reduction of EE at N/P 20/1 comparing to that at 15/1. For cationic polymers, the EE% for PEI is pervasively higher than that from chitosan. A comparable EE% was observed between spermine-EEPG NPs and PEI-EEPG NPs. For cell-penetrating peptides, besides of Transportan, K9/KALA/Penetratin-paired NPs showed enhanced EE (> 80%) after increasing N/P ratio to 20/1. In category of cationic lipids, the positive correlation between N/P ratio and EE% was observed across all tested molecules, with Lipid 2/Lipid 10-paired NPs excelling by achieving an EE > 80%, outperforming other lipids.

From a pharmaceutical standpoint, to maximize the siRNA encapsulation with low consumption of biomaterials, we chose the nanosystems with EE% over 80%, that included Spermine/PEI800/PEI25K/Lipid 2/Lipid 10/K9/KALA-EEPG NPs, for following screening.

**Biocompatibility of the nanoformulations**

With a satisfied siRNA encapsulation efficiency, we then sought to evaluate the in vitro biocompatibility of afore-filtered nanosystems. Considering the macrophages tropism capability from the NPs, which is inherently generated by the addition of EEPG[6], the biocompatibility of promising formulations was tested on human monocytic cells THP-1 and RAW 264.7 murine macrophages cells. THP-1 monocytes were pre-stimulated by 100 ng mL$^{-1}$ phorbol myristate acetate (PMA) for 24 h to obtain macrophages. Afterwards, THP-1 derived macrophages and

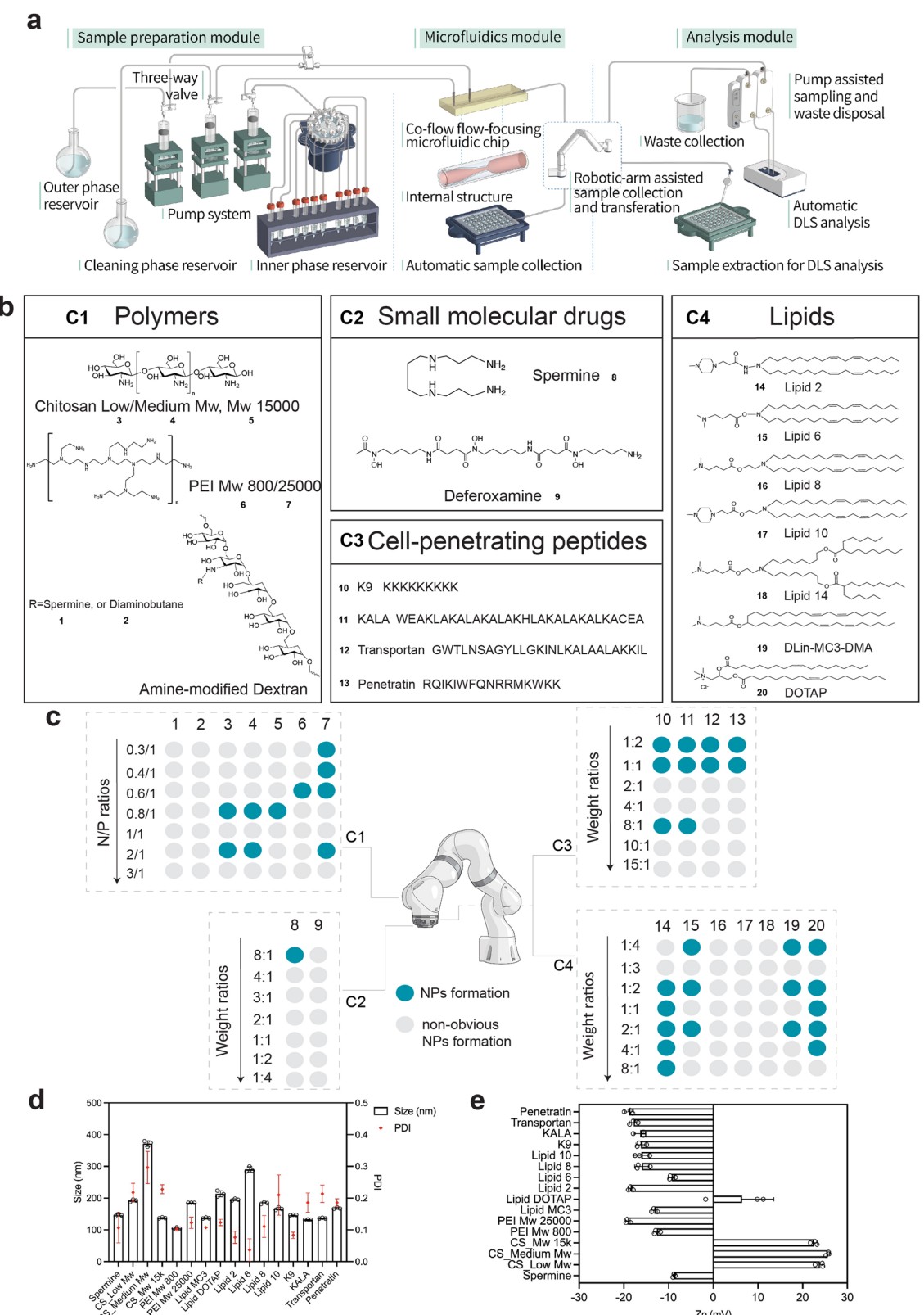

**Fig. 2 | "Robot assisted" microfluidic preparation and screening of EEPG-paired nanoparticles. a** Overview of the self-designed automated microfluidics system for formulation preparation; **b** Chemical structures and names of 20 cationic compounds for constructing EEPG-based NPs in a combinatorial manner. C1-C4, Category 1-4; **c** Graphic illustration of the initial screening pool, where each category of cationic compounds was set 7 different weight ratios (w/w) or nitrogen/phosphate

(N/P) ratios, and blue dot indicated the formation of NPs, whereas grey dot indicated the weak interaction of the two compartments without pervasive formation of NPs; **d** Representative nanoformulations with optimal hydrodynamic diameter and polydispersity (PDI) ($n = 3$, replicates), **e** Zeta ($\zeta$)-potential value of representative 16 nanoformulations ($n = 3$, replicates). Data are presented as mean ± SD, $n = 3$. Source data are provided as a Source Data file.

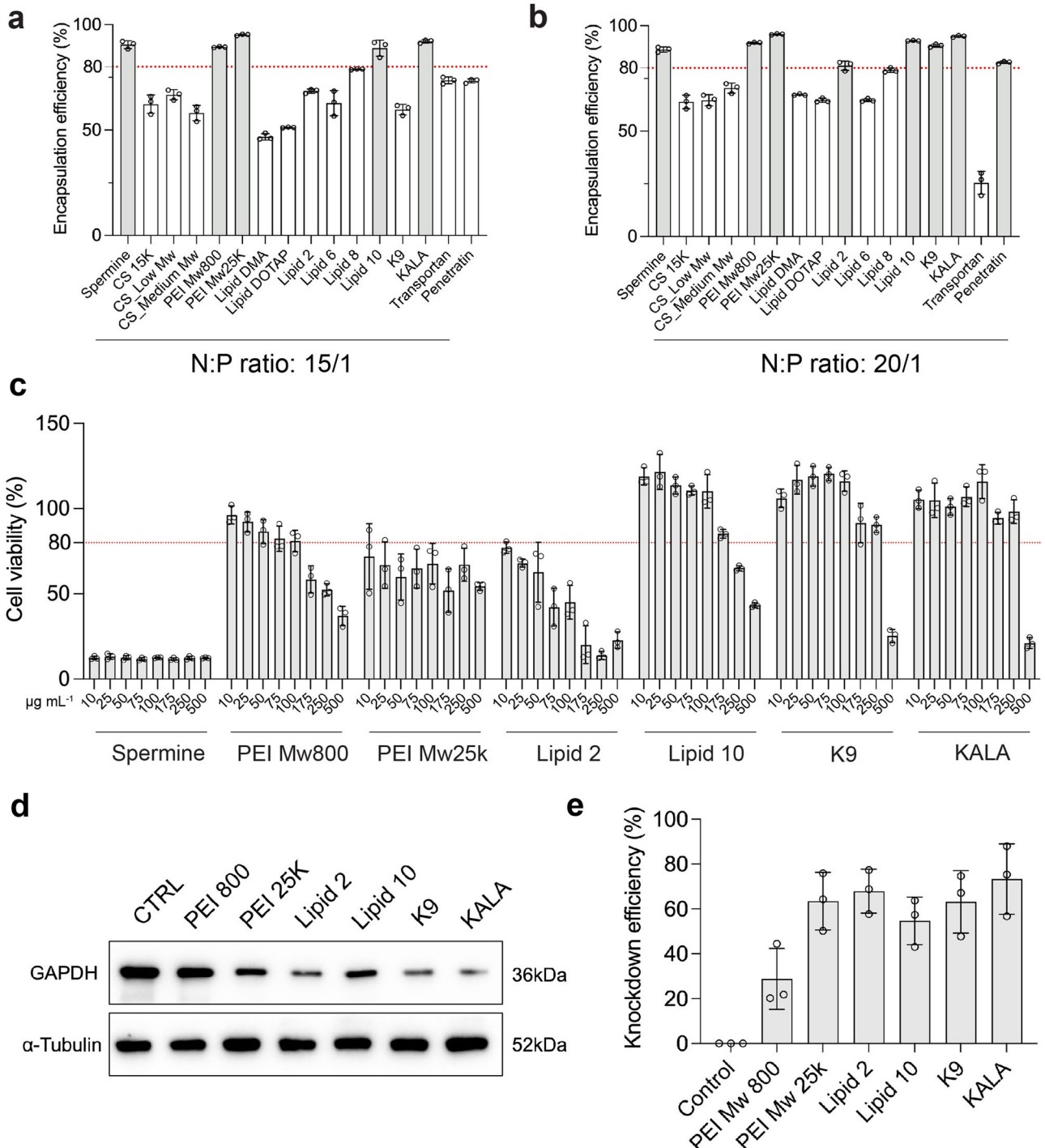

**Fig. 3 | Step-wise screening of EEPG-paired nanoparticles for potential siRNA delivery.** RiboGreen assay was adopted to determine the encapsulation efficiency of siRNA in each cationic compound-paired nanoparticles, the nitrogen/phosphate (N/P) ratios were set at 15/1 (**a**) and 20/1 (**b**) respectively; **c** THP-1 cytotoxicity evaluation of 7 nanosystems including spermine-EEPG, PEI800-EEPG, PEI25K-EEPG, Lipid 2-EEPG, Lipid 10-EEPG, K9-EEPG and KALA-EEPG, the concentrations were correspondence to the total NPs concentration; **d, e** Western blotting analysis of GAPDH expression in THP-1 derived macrophages. α-Tubulin was used as an internal control. Data are presented as mean ± SD with triplicates ($n = 3$). Source data are provided as a Source Data file.

RAW 264.7 cells were separately incubated with 10–500 μg mL$^{-1}$ of various EEPG-paired NPs for 48 h. We have previously confirmed the satisfied biocompatibility from EEPG towards different macrophages cell lines[6]. Upon the ionotropic gelation with different cationic compounds to form NPs, spermine-EEPG NPs showed the most significant loss of cell viability even at lowest doses (10 μg mL$^{-1}$). Low-molecular-weight PEI (PEI, Mw = 800) based NPs (PEI800-EEPG) showed an overall better cellular tolerability compared to their high-molecular-

weight counterparts (PEI25K-EEPG, IC50 = 298 μg mL$^{-1}$ *vs.* 227 μg mL$^{-1}$), which is consistent with previous reports, where the molecular weight of PEI is positively associated with cytotoxicity[19]. For cationic lipids, Lipid 2 showed substantial loss of cell viability (< 50%) at dosages higher than 75 μg mL$^{-1}$, where Lipid 10 only showed significantly cytotoxic at dosages over than 250 μg mL$^{-1}$. Compared to other categories, K9-EEPG and KALA-EEPG NPs showed an altered, yet relatively satisfied biocompatibility towards macrophages, where a significant

cellular toxicity was only observed at highest dosages conditions (500 μg mL⁻¹, Fig. 3c). In addition, the biocompatibility of different EEPG-paired NPs in RAW 264.7 cells were pervasively higher than that in THP-1 cells (Supplementary Fig. 4). Considering the inherent targeting capability from EEPG towards a C-type lectin-1 receptor, Dectin-1[20], the difference may be partly attributed to the distinct Dectin-1 expression profile from different cells, where RAW 264.7 is known as a Dectin-1⁻ cell line and THP-1 cells are Dectin-1⁺[21,22]. The biased biocompatibility of EEPG-based NPs also partly confirmed the preserved Dectin-1 targeting capability from EEPG after fabricating into NPs. Nevertheless, with these results, we excluded spermine-EEPG NPs for further screening to assess their efficacy on gene silencing.

## Gene silencing efficacy of nanoformulations

With the identification of in vitro biocompatibility, finally, we evaluated the gene silencing efficacy of afore-filtered nanosystems. In total 6 EEPG-based NPs were selected, including PEI800-EEPG, PEI25K-EEPG, Lipid 2-EEPG, Lipid 10-EEPG, K9-EEPG and KALA-EEPG. GAPDH-counteracted siRNA (siGAPDH) was adopted as a model siRNA to be encapsulated. Due to the Dectin-1 targeting efficiency from EEPG[23], Dectin-1⁺ human monocytic cells, THP-1, were utilized to test the knockdown efficiency of different formulations[6]. The working concentration of siGAPDH was kept at 50 nM in all tested NPs, after incubating with cells for 48 h, the silencing efficiency towards GAPDH was quantitatively evaluated by western blotting analysis, with α-Tubulin serving as internal control (Fig. 3d). The western blotting results indicated a superior gene silencing efficacy from siGAPDH-loaded PEI25K-EEPG NPs comparing to PEI800-EEPG. For cationic lipid-paired NPs, despite Lipid 2 and Lipid 10 exhibited almost identical chemical composition and molecular structure, we observed the gene silencing efficiency from Lipid 2-EEPG NPs was slightly higher than that from Lipid 10-EEPG NPs, despite no statistical significance was detected. Similar to cationic lipids, cationic peptides-paired NPs also exhibited potent gene silencing efficacy. Similarly, in a same siGAPDH concentration (50 nM), K9-EEPG and KALA-EEPG NPs showed slightly difference in regarding to gene knockdown efficiency, and among which, KALA-EEPG NPs exhibited the highest knockdown levels (73%) at the siRNA concentration of 50 nM (Fig. 3e).

In summary, we conducted a step-wise screening of formulation optimization, including NP-formation, siRNA encapsulation, biocompatibility and in vitro knockdown efficiency. The generated pairing NPs composed by EEPG and cell-penetrating peptide KALA exhibited preferred properties in all screening steps and therefore will be further tested for biological applications in the following experiments.

## Computational-assisted nanoformulation screening

Besides of the identified promising NPs, we sought to understand the pattern beneath the selection process. However, considering the limited size of the dataset, to design a robust algorithm for cationic compounds prediction by conventional cheminformatics method is challenging. Our goal was rather to extract potential scientific understanding from the screening process and to provide perspectives for interpretation methods, as this interpretability holds particular value for medical applications[12,24]. We first produced a classification model in the form of a decision tree (DT) to summarize the screening process (Fig. 4a). Dark blue squares and grey squares separately indicated the positive and negative compound sets for each screening, and the light blue diamond described the screening criteria. For NPs formation, in a typical scenario of polyelectrolytes-based gelation, the binding and aggregation is usually prone to be observed in an electro-neutral manner[25]. However, in the current study, the phenomenon can be divided into 3 categories: (1) cationic reagents that did not form stable NPs with EEPG at any concentration (samples 15, 16, 17, and 20); (2) cationic compounds formed colloidal stable NPs only at N/P ratios significantly deviating from unity (< 0.1 or >10, samples 14, 18, and 19); and (3) cationic reagents that can steadily form into NPs (samples 1–13, Fig. 4b, Supplementary Fig. 5a). We first thought whether a molecular descriptor (MolDes) based quantitative structure-activity relationship (QSAR) model can provide potential explanation. MolDes of each cationic compound was generated by alvaDesc (Methods), which were sequentially adopted for logistic regression based feature selection with binary formation of NPs as label[26]. However, even for MolDes with the lowest p-value (indicating the MolDes holds the most significant association with the NPs formation process)[27], its corresponding receiver operating characteristic (ROC) curve only showed an area under the curve (AUC, indicating the performance of the MolDes in distinguishing between the positive and negative classes) of 0.85 (Supplementary Fig. 5b, c), which limited the extraction of scientific understanding from the current model. To gain better insights, we turned to short (1 ns) molecular dynamics (MD) simulations to quantify non-covalent interaction potentials among EEPG and different cationic compounds on the basis of electrostatic interaction (Coulombs potential) and electronically neutral van der Waals interaction (Lennard-Jones, LJ potential)[28]. For negative subsets (sample 14–20, Supplementary Fig. 5a), the inter-Coulomb potential between EEPG and counter-reagents consistently approached or even exceeded 0, whereas the corresponding values for positive group were typically below 0 (Fig. 4c), suggesting minimal contribution from electrostatic interaction in cationic reagents that did not form stable NPs with EEPG. We further evaluated the difference between the cationic compounds that did not form stable NPs with EEPG at any concentration (samples 15, 16, 17, and 20), and cationic compounds formed colloidal stable NPs only at N/P ratios significantly deviating from unity (< 0.1 or >10, samples 14, 18, and 19, Supplementary Fig. 5a). Upon simulation, the inner LJ potential of EEPG in non-electronic binding groups were significantly lower, resembling a self-aggregation tendency from EEPG (Fig. 4d), partly suggesting the NPs formation may be attributed to the self-aggregation from EEPG through non-polar interaction. Overall, MD simulations may be adopted as a valuable supplementary tool for initial evaluations and insights into the inotropic gelation based NPs formation process, which is mainly reflected by observing the Coulomb's potential during the simulation.

Next, we try to elucidate the critical factors in governing the siRNA encapsulation efficiency (EE%). Considering that the data generated from MD simulations may offer supplementary insights beyond the conventional MolDes computed by alvaDesc, we combined the data generated from MD with the conventional MolDes calculated by alvaDesc, which were further subjected to feature selection with EE% as labels. Feature selection analysis indicated that the average LJ potential between EEPG and counter-reagent had the highest r-square value ($r^2 = 0.7$, Fig. 4e), which is substantially higher than those conventionally calculated MolDes[29]. Considering the LJ potential is correlated with the distance between two molecules, we hypothesize this correlation might be attributable to steric repulsion exerted by the cationic reagents during siRNA encapsulation[30], whereas the stronger interaction between EEPG and the cationic compound might induce spatial hindrance for siRNA residence, negatively affecting siRNA encapsulation. To confirm our hypothesis, we separately visualized binding of EEPG with KALA and Transportan. These two peptides were selected for comparison based on their similar molecular weight, hydrophobicity (LogP) and surface charges, yet markedly different EE% (KALA, EE% = 95%; Transportan, EE% = 25%). Different from EEPG-KALA complex, where KALA was mobilized in one tail of EEPG in a flexible manner, Transportan was enveloped by EEPG in a more compact structure (Fig. 4f). This was also partly reflected as EEPG-Transportan complex formed over ten times more close contacts (< 6 Å) than the EEPG-KALA complex (Supplementary Fig. 5d).

In our sequential investigation into the factors contributing to the observed biocompatibility of various formulations, we employed a

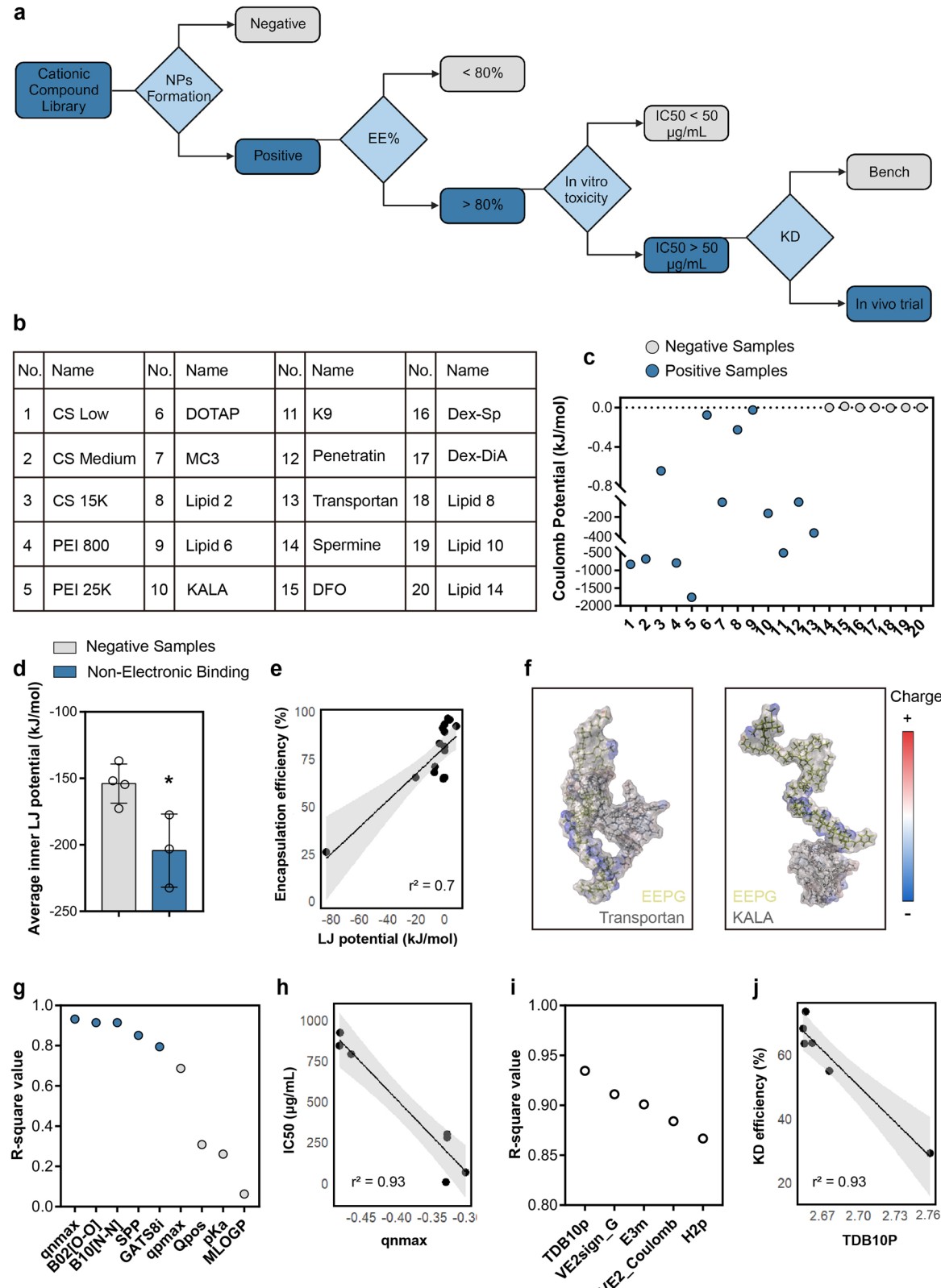

similar independent feature selection methodology, utilizing the IC50 values as labels and the MolDes of each cationic compound as variables. Multiple studies have adopted MolDes to identify association between physicochemical properties and toxicity. Notably, molecular weight (Mw), lipophilicity (log P), and basicity (pKa of strongest base) have been identified in multiple pharmaceutical studies[31,32]. However, the current results potentially suggested the significance of number

and topological location of negatively charged atoms in mediating the promiscuity of a compound. This was evidenced as a MolDes, qnmax, which represented the maximum negative charge of the compound, potentially showed highest coefficient of determination ($r^2 = 0.93$, Fig. 4g, h, Supplementary Table 7). Similarly, the third leading MolDes, SPP, was also correlated with the absolute difference between the maximum negative charges and maximum positive charges

**Fig. 4 | Computational assisted step-wise interpretation of formulation screening process. a** A decision tree-like scheme was built up to summarize the screening process, where dark blue squares represent the positive sets identified at each stage of screening, while grey squares denote the negative sets. Additionally, light blue diamonds are used to delineate the specific criteria applied at each screening juncture. KD (knockdown); **b** A summarized table to show the tested cationic compounds in the computational analysis; **c** Quantitative analysis of Coulombs potential by 1 ns molecular dynamics (MD) simulations with different EEPG pairings; **d** Quantitative analysis of average inner Lennard-Jones (LJ) potential from EEPG by 1 ns molecular dynamics (MD) simulations from cationic compounds that did not form stable NPs with EEPG at any concentration (Negative Samples, samples 15, 16, 17, and 20), and cationic compounds formed colloidal stable NPs only at N/P ratios significantly deviating from unity (Non-Electronic Binding, samples 14, 18, and 19). Data are presented as mean ± SD. Statistical analysis was conducted by two-tailed Student's $t$-test, where $*p = 0.0248$; **e** Linear correlation between encapsulation efficiency and average LJ potential between EEPG and the paired cationic compound; **f** Molecular dynamics (MD) simulation was performed to reflect the binding structure of Transportan-EEPG and KALA-EEPG; **g** Coefficient of determination of different MolDes in correlation to IC50 of each formulation as calculated by feature selection. Blue circles indicated the top 5 MolDes with the highest Pearsons' correlation coefficient and light circles indicated MolDes which were previously reported to be highly correlated to the cytotoxicity of a compound; **h** Linear correlation between IC50 of the formulation and qnmax value from cationic compound; **i** Squared Pearsons' correlation coefficient of top 5 MolDes in correlation to knockdown efficiency of each formulation; **j** Linear correlation between knockdown efficiency of the formulation and TDB10p value from cationic compound. Grey area in **e**, **h** and **j** represent 95% confidence interval. Source data are provided as a Source Data file.

(Supplementary Table 7)[33]. Other leading MolDes included B02[O-O] and B10[N-N], which separately represented the presence/absence of O-O (N-N) at topological distance 2/10 (Supplementary Table 7)[34]. Despite the hydrophobicity and molecular charges, mostly positive, have been widely acknowledged for assessing biocompatibility of a compound, from the above, we suggested that the distribution and ratio between negative and positive parts of a compound may also occupy a significant place in mediating the biocompatibility of the molecules.

In our last pursuit to elucidate the relationship between the chemical properties of cationic counterparts and the gene knockdown (KD) efficacy of the resultant NPs, feature selection was conducted with KD efficiency of each formulation as the label and MolDes of each cationic compounds as variables. Coefficients of determination of each MolDes is adopted for evaluating the impact of each MolDes, as it represents the proportion of the variance for KD efficiency to be affected by a specific MolDes. We noted that among the top 5 MolDes with the highest coefficient of determination, 3 of them (TDB10p, E3m and H2p) belonged to 3D MolDes[27,35,36], with TDB10p exhibited the highest $r^2$ value of 0.93 (Fig. 4i–j). These MolDes typically encoded both the 3D-geometrical information and chemical information with different weighting scheme such as polarizability (TDB10p, H2p) and mass (E3m) from the atoms. The detailed description of each MolDes is shown in Supplementary Table 7. Previous studies demonstrated the transfection efficiency from different cationic polyelectrolytes, even with the identical chemical compositions, were largely affected by their 3D orientation[37,38]. Thus, conventional MolDes with clear chemical or physical attributes, despite their better interpretability, may not be enough for understanding the divergent phenomenon observed during the experiments[39], as partly confirmed by the unsatisfied $r^2$ value from MolDes describing hydrophobicity, polar surface area and positive charges (Supplementary Fig. 5e, Supplementary Table 7). Therefore, the current results highlighted the potential importance of 3D geometry of a cationic compound in affecting the transfection efficiency of the fabricated nanosystem.

### Identification and characterization of GluCARDIA NPs

With a systematic evaluation of key parameters for NPs formation and siRNA delivery, we identified pairing NPs between EEPG and KALA to form siRNA loaded NPs. Considering the precise Dectin-1 targeting efficiency from EEPG[6], along with the abnormally activated expression of Dectin-1 in injured myocardium upon myocardial ischemic/reperfusion (IR) injury[40], as a proof of concept, we determined to test the therapeutic potency of identified KALA-EEPG NPs for ameliorating cardiac reperfusion injuries, and the NPs were further termed as β-Glucan based Combinatory Assessed delivery system for Reperfusion Damage Inhibition and Aid (GluCARDIA) NPs. Prior to this, the morphology of bare GluCARDIA, as well as siRNA-loaded GluCARDIA NPs was analyzed by transmission electron microscopy (TEM, Fig. 5a, b). TEM images show that both of them showed a homogenous size and

morphology, which is in good accordance with the physicochemical parameters evaluated by DLS (Fig. 5c). The mean size of GluCARDIA-siRNA NPs was slightly increased from 132 ± 0.6 nm to 137 ± 1.6 nm due to the incorporation of siRNA, whereas siRNA incorporated GluCARDIA NPs still exhibited a narrow polydispersity index (PDI) of 0.12.

### Cytosolic delivery of siRNA

Next, we investigated the GluCARDIA NPs cytosolic delivery of siRNA, which is critical for RNAi machinery function[6,41]. To this end, the fluorescence dye labelled siRNA (Cy5-siRNA) encapsulated GluCARDIA NPs (GluCARDIA-Cy5-siRNA NPs) were incubated with THP-1 cells for 1 h, 2 h and 6 h (Fig. 5d), and lysotracker was co-stained to unravel the acidic endosomes/lysosomes intracellular compartments during endocytosis. After 1 h incubation, GluCARDIA-Cy5-siRNA NPs was observed, whereas a major colocalization between NPs and lysotracker was measured, indicating their major entrapment in endosomes/lysosomes organelles. Followed by increasing the incubation time, a gradual disassociation between Cy5-siRNA and lysotracker was observed in a time-dependent manner, suggesting the efficient endosome escape of siRNA facilitated by GluCARDIA NPs. This process was further quantified by evaluating the Manders' colocalization coefficient (MCC) between lysotracker and GluCARDIA-Cy5-siRNA NPs[42], where the MCC was gradually decreased from 0.6 to 0.2, suggesting the time-dependent endosome escape process mediated by GluCARDIA NPs (Fig. 5e). Our previous studies confirmed that EEPG with a pH-dependent membrane-lytic capability, upon acidification, EEPG can destabilize the membrane and facilitate siRNA delivery into cytosol during the maturation process of endosomes[6]. Overall, these results indicate the efficient cellular uptake and endosomal escape properties of GluCARIDIA-based siRNA delivery system.

### Targeting Dectin-1⁺ macrophages with gene silencing

One major advantage of polysaccharide-based nanosystems is their innate targeting capability towards specific lectins, enabling a targeted delivery without subsequent modifications[7]. β-glucans, the building block for EEPG, are inherent and precise ligands for Dectin-1[40,43]. Thus, we sequentially tested the Dectin-1 targeting capability from GluCARDIA NPs. We compared the cellular interaction of GluCARDIA NPs on different macrophage cell lines, including Dectin-1⁺ THP-1 and Dectin-1⁻ RAW 264.7 macrophages[22,44]. First, the cellular uptake of GluCARDIA-Cy5-siRNA NPs in different cell-lines was quantitatively evaluated by flow cytometry analysis. As expected, free Cy5-siRNA barely showed any cellular uptake in both cell lines, and the obvious cellular uptake were only observed when Cy5-siRNA were incorporated in GluCARDIA NPs (Supplementary Fig. 6-7). Moreover, compared to RAW 264.7 cells, THP-1 showed a statistically significant higher cellular uptake towards GluCARDIA as shown by the positive cell percentages differences (Fig. 5f).

We further evaluated whether this targeting efficiency is associated with gene knockdown efficiency in macrophages with different

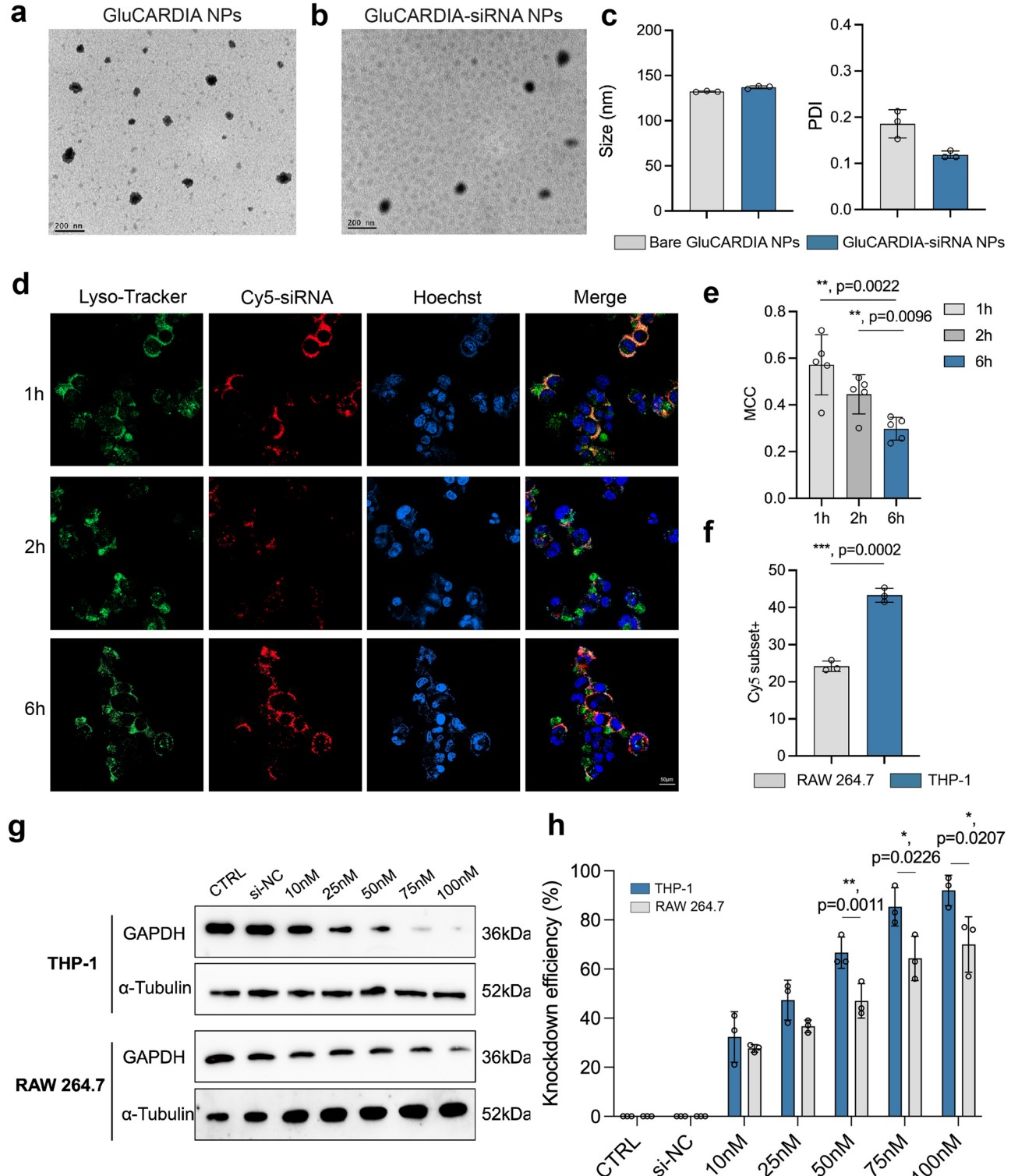

**Fig. 5 | Characterization and in vitro evaluation of GluCARDIA NPs.** Morphology of bare GluCARDIA NPs (**a**) and GluCARDIA-siRNA NPs (**b**) were observed by TEM analysis; **c** DLS analysis of GluCARDIA NPs and GluCARDIA-siRNA NPs to evaluate their corresponding hydrodynamic diameter and PDI (*n* = 3, replicates); **d** Time-dependent confocal microscopy images of GluCARDIA-Cy5 siRNA in THP-1 macrophages delineating the intracellular translocation of Cy5-siRNA; **e** Mander's Correlation Coefficient (MCC) between Cy5-siRNA and Lyso-Tracker at different time-points were quantified to show the time-dependent endosome escape of siRNA (*n* = 5); **f** Flow cytometry results of GluCARDIA Cy5-siRNA NPs uptake in RAW 264.7 and THP-1 cells (*n* = 3, replicates); Western blotting images (**g**) and corresponding quantitative analysis of GAPDH expression in THP-1 or RAW 264.7 cells after treated with GluCARDIA-siGAPDH NPs at different siRNA concentrations (**h**), (*n* = 3, replicates). α-Tubulin was used as an internal control. Data are presented as mean ± SD with at least triplicates. Statistical analysis was conducted by two-tailed Student's *t*-test, where *\*p* < 0.05; *\*\*p* < 0.01; *\*\*\*p* < 0.001. Source data are provided as a Source Data file.

Dectin-1 expression pattern. To this end, siGAPDH with varied dosage (ranging from 10 nM to 100 nM) were loaded into GluCARDIA NPs (GluCARDIA-siGAPDH NPs), followed by incubating with THP-1 or RAW 264.7 cells for 48 h. Cells from different groups were then subjected to western blotting analysis to evaluate the gene knockdown efficiency (Fig. 5g, h). The results showed a superior gene silencing efficacy from GluCARDIA-siGAPDH NPs in THP-1 cells compared to that from RAW264.7 cells, indicating the enhanced genetic cargo delivery into Dectin-1+ cells.

To exclude the possibility that the promoted cellular interaction and the concomitantly achieved higher gene silencing efficacy in Dectin-1+ macrophages from GluCARDIA NPs were endowed by the addition of KALA, but rather generated by the incorporation of EEPG. As a side-by-side comparison, we compared the gene silencing ability of different siGAPDH encapsulated EEPG-based NPs (namely PEI800-EEPG, PEI25K-EEPG, Lipid 2-EEPG, Lipid 10-EEPG, K9-EEPG, KALA-EEPG) in both THP-1 cells (Fig. 3d) and RAW 264.7 cells (Supplementary Fig. 8). The results suggested under the same siRNA dosage (50 nM), the gene knockdown efficiency from the same NPs was pervasively higher in THP-1 cells comparing to RAW 264.7 cells, confirming the promoted gene silencing was more likely to be correlated to the EEPG but not from the cationic counterparts.

Overall, we confirmed that GluCARDIA NPs have a higher cellular uptake as well as gene knockdown efficiency in Dectin-1+ macrophages comparing to their Dectin-1- counterparts, partially indicating the Dectin-1 correlated gene silencing from GluCARDIA in vitro.

## In vivo targeting to cardiac IR

After the in vitro results, we sought to investigate whether this targeting efficiency is also valid in vivo and its potential biological benefits in promoting the cardiac accumulation of GluCARDIA NPs in mice with myocardial reperfusion injury. Prior to this, the biosafety property was evaluated with intravenous (i.v.) administration of GluCARDIA NPs at the dosage of 4.8 mg kg$^{-1}$ (pre-determined by the potential siRNA dosage of 2.64 μg per mouse). After 48 h, the serum levels of alanine aminotransferase (ALT), aspartate aminotransferase (AST), and total bilirubin (TBIL) were quantified to assess the liver function (Fig. 6a–c). Negligible differences were shown within two groups, suggesting that GluCARDIA NPs were well-tolerated without undesirable liver toxicity in short term.

Next, we evaluated the biodistribution of GluCARDIA NPs in a well-established preclinical IR murine model[8]. To this end, indocyanine green labelled GluCARDIA NPs (ICG-GluCARDIA NPs) was synthesized through a previous established protocol for fluorescent imaging purposes[8]. As a comparison, we applied dextran, a type of α-glucan, served as a negative polysaccharide control without Dectin-1 binding affinity[45]. ICG was encapsulated in acetalated dextran NPs according to previous protocol to fabricate ICG-Dextran NPs[46]. Mice were subjected to 50 min of ischemia followed by 24 h of reperfusion, afterwards, ICG-GluCARDIA or ICG-Dextran NPs was i.v. injected for further analysis. Major organs including heart, lung, liver, spleen and kidney were harvested at 8 h post-injection for fluorescence imaging. The cardiac accumulation of different NPs was calculated based on a previously established term, heart-targeting index (HTI, HTI = heart fluorescence emission/liver fluorescence emission)[47]. The results show a clear statistically significant difference in cardiac accumulation between ICG-GluCARDIA NPs and ICG-Dextran NPs (Fig. 6d), with a significant higher HTI observed from mice treated with ICG-GluCARDIA (Fig. 6e), suggesting the feasibility of adopting GluCARDIA NPs for promoted cardiac accumulation post-IR.

Despite the substantial lower cardiac accumulation comparing to hepatic accumulation, we further determined to evaluate its potential impact on gene silencing, and the knockdown efficiency was evaluated by western blotting analysis (Supplementary Fig. 9). Results showed that while the overall accumulation of GluCARDIA NPs in cardiac tissue is significantly lower than that in liver, the gene knockdown efficiency was still comparable to that observed in hepatic tissue. This might be explained by the "siRNA saturation" effect as described by Lieberman et al., as only marginal amount of the siRNA (~ $10^{-9}$ pmol) is needed to achieve a maximal gene knockdown efficiency, and further increasing of the siRNA showed no significant difference in terms of gene knockdown yield[48]. Therefore, despite the relative cardiac accumulation of GluCARDIA is still substantially lower than that in liver, the currently developed nanosystem can significantly promote the cardiac accumulation and sequentially achieve the envisioned design. Meanwhile, RNA-seq suggested the administration of GluCARDIA-siIRF3 exhibited minimal impact on major liver function in the context of myocardial IR, indicative of a satisfied biosafety property for the current therapeutic strategy (Supplementary Fig. 10, Supplementary Data 3).

To further identify the potential association between GluCARDIA NPs with other types of immune cells in injured myocardium, we conducted double-immunofluorescence staining between FITC labelled GluCARDIA NPs (FITC-GluCARDIA) and Dectin-1 (Figs. 6f) or Ly-6G (neutrophil marker, Fig. 6g) in the mice underwent IR injury[6,8]. The results show a major co-localization between Dectin-1 and FITC-GluCARDIA NPs, with negligible correlation observed from Ly-6G. Further quantitative assessment using MCC calculation revealed a value of 0.96 between GluCARDIA and Dectin-1, which was significantly higher compared to that with Ly6G (0.38, Fig. 6h).

A more comprehensive investigation on the potential interaction between the administrated NPs with different immune cells were evaluated by flow cytometry analysis. Three groups were subjected to FACS analysis: fluorescein isothiocyanate (FITC)-labelled GluCARDIA NPs; FITC-labelled Dextran NPs (utilized as a negative control due to its absence of Dectin-1 targeting affinity); and FITC-labelled SM-102 LNPs (utilized as a comparative standard for commercially available NPs in gene delivery). Mice were subjected to 50 min of ischemia followed by 24 h of reperfusion, afterwards, FITC-labelled GluCARDIA NPs/Dextran NPs/SM-102 LNPs was i.v. injected for further analysis. 8 h post-injection, organs including heart and liver were harvested for flow cytometry analysis (Supplementary Fig. 11a). In the lesioned myocardium, GluCARDIA NPs mainly co-localized with monocytic cells, which was substantially higher comparing to Dextran NPs and SM-102 LNPs (Supplementary Fig. 11b). Notably, compared with Dextran NPs and SM-102 LNPs, GluCARDIA NPs also exhibited an increased recognition by dendritic cells, which are also known as major Dectin-1+ cellular subsets upon sterile inflammation[49], while exhibiting a comparable cellular uptake with Dectin-1- cells such as neutrophils[40].

Regarding to the potential cellular interaction with hepatic immune cells, despite the major colocalization was still observed in monocytic cells, in contrary to the enhanced monocytic targeting efficiency in lesioned myocardium, GluCARDIA NPs exhibited a similar monocytic internalization efficiency comparing to SM-102 LNPs (Supplementary Fig. 11c). This may be attributed to the restrained Dectin-1 expression profile in hepatic macrophages upon sterile injury[6]. In conclusion, here we have demonstrated that in the setting of myocardial IR, our designed GluCARDIA NPs are primarily recognized by monocytic cells, significantly exceeding its uptake in other immune cells. Moreover, GluCARDIA NPs exhibited substantially higher targeting efficiency towards cardiac monocytes comparing to α-polysaccharides (dextran) based NPs and commercialized benchmark NPs (SM-102 LNPs), offering a promise delivery system for efficient cardiac gene delivery post IR.

## Gene silencing after cardiac IR

We further aimed to construct GluCARDIA NPs loaded with functional siRNA for potential cardiac therapeutic interventions. Previous studies indicated cardiac injury provoked interferon regulatory factor 3 (IRF3) activation in recruited macrophages in infarct myocardium,

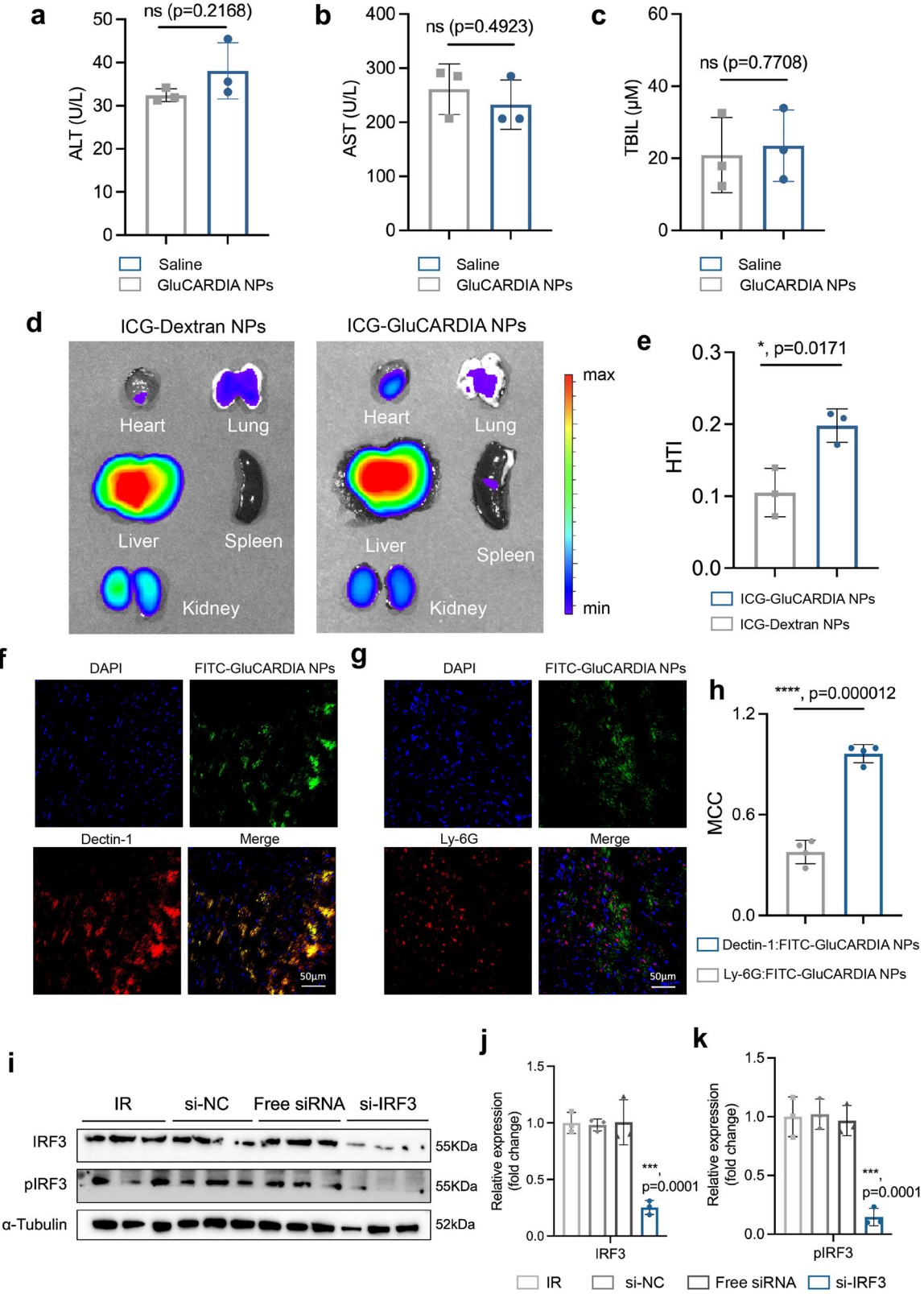

potentially leading to adverse ventricular remodeling and poor prognosis[50], therefore, we hypothesized that siIRF3-loaded GluCARDIA NPs (GluCARDIA-siIRF3) yield a beneficial effect for cardiac protection after IR. Mice were subjected to 50 min of ischemia followed by different period of reperfusion, and GluCARDIA loaded with non-targeting siRNA controls (GluCARDIA-siNC), free siIRF3 and GluCARDIA-siIRF3 were i.v. injected at 30 min, 24 h and 72 h post-

reperfusion at the siRNA dosage of $0.1\,mg\,kg^{-1}$. We evaluated the protein expression level of IRF3 and phosphorylated IRF-3 (p-IRF3) after therapeutic intervention. To this end, injured left ventricular (LV) tissues were harvested at day 4 post-reperfusion, and then subjected to western blotting analysis. We detected marked decrease of IRF3 and p-IRF3 (Fig. 6i–k) level in LV tissue solely from GluCARDIA-siIRF3, and this phenomenon was not observed from GluCARDIA-siNC or free

**Fig. 6 | Biodistribution and gene silencing efficacy of GluCARDIA NPs in murine cardiac IR model. a–c** Short term liver toxicity from GluCARDIA NPs upon intravenous injection were evaluated by serum alanine aminotransferase (ALT), aspartate aminotransferase (AST) and total bilirubin (TBIL) levels ($n = 3$, replicates); **d** Representative fluorescent mapping of ICG-Dextran NPs and ICG-GluCARDIA NPs in major organs at 24 h post-IR. **e** Heart targeting index (HTI) was quantified to compare the heart accumulation efficacy of ICG-Dextran NPs and ICG-GluCARDIA NPs ($n = 3$, replicates). **f–g** Representative dual-immunofluorescence staining of ICG-GluCARDIA NPs and Dectin-1 (**f**) or ICG-GluCARDIA NPs and Ly-6G (**g**) at 24 h post-IR; **h**, Mander's Correlation Coefficient (MCC) between ICG-GluCARDIA and Dectin-1 or ICG-GluCARDIA and Ly6G were quantified to confirm the cellular interaction of GluCARDIA NPs ($n = 4$, replicates); **i–k** Western blotting analysis of IRF3 and pIRF3 expression in injured heart tissues. α-Tubulin was used as an internal control ($n = 3$, replicates). Data are presented as mean ± SD with at least triplicates. Statistical analysis was conducted by two-tailed Student's *t*-test or one-way ANOVA with Tukey's correction, where *$p < 0.05$; **$p < 0.01$; ***$p < 0.001$ and ****$p < 0.0001$. Source data are provided as a Source Data file.

siIRF3 administration, indicative to a potent in vivo gene silencing efficacy from GluCARDIA NPs.

### Protection of heart after myocardial IR

With identifying the satisfied genetic silencing efficacy in vivo, we finally sought to determine the functional significance of GluCARDIA-siIRF3. To this end, mice underwent same IR scenario were i.v. injected with GluCARDIA-siNC and GluCARDIA-siIRF3 at 30 min, 24 h and 72 h post-reperfusion at the siRNA dosage of 0.1 mg kg⁻¹ (Fig. 7a). The 24 h post the reperfusion, a reduced terminal-deoxynucleotidyl transferase mediated nick end labeling (TUNEL) cells were observed from GluCARDIA-siIRF3 group, indicative of a potential cardioprotective effects of the current treatment (Supplementary Fig. 13). Furthermore, we assessed the transcription levels of key regulatory genes in each group, including TGF-β, interferon-β, and caspase-3 to reflect the fibrosis, immune responses, and cellular apoptosis status following myocardial IR. Comparing to GluCARDIA-siNC or IR group, marked decreases in transcriptional levels of these genes were noted in the GluCARDIA-siIRF3 group, underscoring the potential beneficial effects of GluCARDIA-siIRF3 treatment following myocardial IR (Supplementary Fig. 14). At day 7 post-reperfusion, RNA-seq analysis was conducted on IR, GluCARDIA-siNC and GluCARDIA-siIRF3 groups to collect a genome-wide expression profiling (Supplementary Data 1). Comparing to IR group, Gene Set Enrichment Analysis (GSEA) identified significant negative association between the GluCARDIA-siIRF3 treatment and signaling pathways related to transforming growth factor β (TGF-β) signaling pathway (Fig. 7b) and dilated cardiomyopathy (Supplementary Fig. 12a)[51], indicating a potentially better prognosis regarding to LV remodeling. Similarly, the GluCARDIA-siIRF3 treatment, comparing to GluCARDIA-siNC group, also inhibited fibrosis progress, promoted cardiac contraction and facilitated the inflammation resolution (Supplementary Fig. 12b–e)[52], suggesting the potentially beneficial effects from IRF3 silencing at the initial stages of myocardial IR.

Echocardiograph (ECG) was also conducted to evaluate the cardiac function from different groups. Follow-up with 7 days of reperfusion, the left ventricular ejection fraction (EF%) improved to 36% with GluCARDIA-siIRF3 treatment, comparing to an EF of 29% from IR group, while GluCARDIA-siNC (29%) showed no beneficial impact on cardiac functions. However, at day 7 post-reperfusion, no statistical difference was observed in regarding to LV end-systolic internal dimension (LVIDs) and LV end-diastolic internal dimension (LVIDd, Fig. 7c, d). Sequentially, the preserved cardiac function by GluCARDIA-siIRF3 was still valid at day 28 post-IR, as reflected by the substantially higher EF comparing to IR group. Meanwhile, GluCARDIA-siIRF3 also ameliorated the LV remodeling as reflected by the significantly reduced LVIDs (Fig. 7e, f). Masson's trichrome staining of heart section from each group was also conducted at day 28 post-IR (Fig.7g), demonstrating a preserved wall thickness (Fig. 7h) and lower percentage of fibrosis in the left ventricle (Fig. 7i) in mice treated with GluCARDIA-siIRF3. Adverse remodeling post myocardial IR injury can lead to cardiomyocytes hypertrophy[53], we observed that mice treated with GluCARDIA-siIRF3 showed considerably decreased myocyte cross-sectional area in comparison to GluCARDIA-siNC or IR (Fig. 7j).

Considering the major accumulation of NPs in liver, we then checked the potential liver damage from different groups. RNA-seq

suggested the administration of GluCARDIA-siIRF3 exhibited minimal impact on major liver function in the context of myocardial IR, indicative of a satisfied biosafety property for the current therapeutic strategy (Supplementary Fig. 10). Furthermore, with four weeks follow-up, liver fibrosis status was evaluated by sirius red staining of liver samples from different groups at end points, and we did not observe significant fibrosis in harvested liver tissues from afore-mentioned three groups (Supplementary Fig. 15), suggesting the GluCARDIA was not overtly toxic when administered to mice over long term.

## Discussion

Utilizing non-viral NPs for targeted gene silencing in organs beyond the liver has garnered growing attention over the years. Given the pervasive occurrence of cardiovascular diseases as the leading cause of death worldwide and the constant emergence of new therapeutic targets without specific small molecular drugs, a rational design of NPs to achieve the cardiac delivery and gene silencing holds significant values[11,54]. In the current study, we applied a strategy to screen and identify a polysaccharide-based nanosystem for achieving cardiac gene silencing in a clinically relevant myocardial IR model. Overall, the proposed strategy to aid the screen process and the identified nanosystem GluCARDIA thereof were featured with the following highlights. First, leveraging the aberrant infiltration of Dectin-1⁺ macrophages in injured myocardium, we selected β-glucan derivative (EEPG) as starting material to endow the fabricated GluCARDIA NPs with inherent targeting capability in a one-step fabrication scheme without further surface functionalization. This "targeting-ligands free" principle of design may simplify the fabrication scheme and reduce batch-to-batch variation, highlighting a unique advantage of using polysaccharides to create non-viral NPs for targeted genetic interventions; second, we conducted a diversity-oriented formulation screening to broaden the scope of ionotropic gelation pairings between polysaccharide and a wide array of different types of molecules. Assisted by the automatic microfluidics platform, formulation screening was conducted in a highly reproducible manner. Corresponding step-wise computational assisted interpretation also provided informative insights in the optimization of a specific property, and the generated knowledge may provide scientific insights towards structure-property relationship interpretation; third, comparing to our previous work, which adopted chitosan-EEPG NPs for cardiac siRNA delivery[6], the currently identified GluCARDIA NPs achieved equivalent gene silencing effects with 33% less siRNA dosage and 70% less NPs mass, highlighting the potency of a diversity-oriented screening for identifying the optimal set of conditions in an efficient and robust manner. These findings provide a practical scheme in identifying polysaccharide-based nanosystem for targeted RNAi in a more efficient and controlled manner, and generated formulation here may guide the development of novel active targeting NPs aimed at enhancing myocardial repair and protection.

Synthesizing NPs in a reproducible, predictable and controllable manner is the pre-condition for successful formulation optimization. Here, we adopted a "robot steers" integral automatic microfluidics system to achieve the envisioned purposes. Compared to the conventional bulk approaches, this automatic microfluidic-based mixing system can precisely tailor the volume of starting materials, leading to

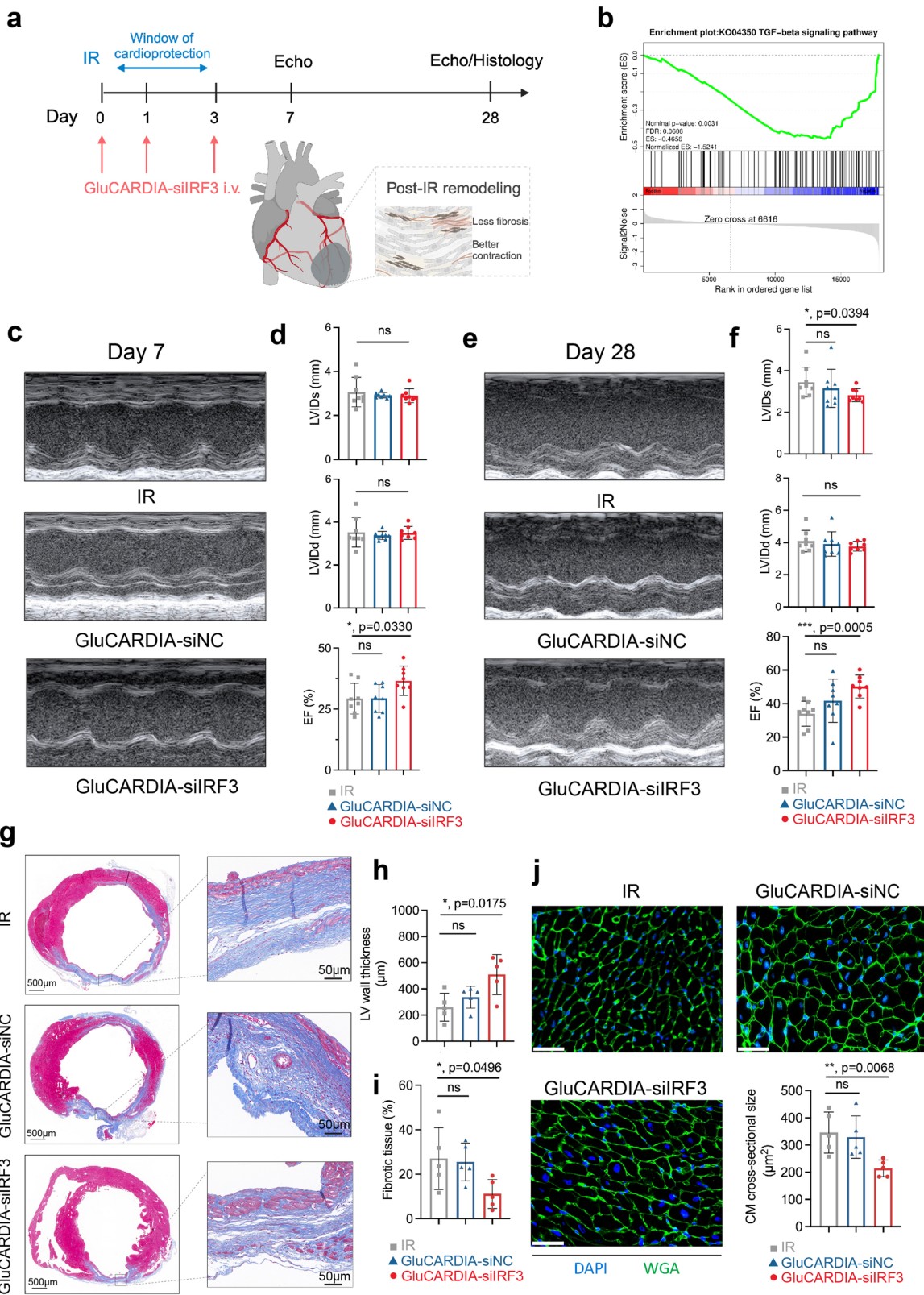

the reduction of batch variations[55]. In this context, we screened in total 161 formulations composed by EEPG and different cationic materials including lipids, polymers, small molecules and peptides. Along with an online characterization system, this innovative microfluidics platform might facilitate the translation of precision nanomedicine into a wider pharmaceutical practice. Evaluating different aspects of a delivery system is crucial to achieve satisfactory therapeutic effects.

Here we performed a step-wise screening process regarding physico-chemical properties, siRNA loading capacity, biocompatibility and in vitro knockdown efficacy, which are key parameters for rational design of nucleic acid delivery vehicles[11]. After 4 rounds of screening, we finally identified the nanosystem GluCARDIA composed by KALA and EEPG held the optimal characters in all aspects. KALA is a cationic amphiphilic peptide that can facilitate siRNA condensation and cellular

**Fig. 7 | GluCARDIA-siIRF3 NPs ameliorated cardiac IR prognosis. a** Schematic illustration of experimental strategy. Mice established with IR injury were intravenously (i.v.) injected with GluCARDIA-siIRF3 at 30 min, 24 h and 72 post-reperfusion; **b** Gene Set Enrichment Analysis (GSEA) between IR and GluCARDIA-siIRF3 at day 7 post-reperfusion indicting a reduced TGF-β signaling pathway activation from GluCARDIA-siIRF3 group, potentially indicating a less fibrosis extent; **c** Representative echocardiographic images at day 7 post-IR; **d** Day 7 echocardiography-based quantification of left ventricular end-systolic internal dimension (LVIDs), left ventricular end-diastolic internal dimension (LVIDd) and ejection fraction (EF) from different groups ($n = 8$); **e** Representative echocardiographic images on day 28; **f** Day 28 echocardiography-based quantification of LVIDs, LVIDd and EF from different groups ($n = 8$); **g** Representative Masson's Trichrome staining of heart tissue sections from different groups at day 28 post-reperfusion; **h** Masson's Trichrome staining-based quantification of left ventricular (LV) wall thickness from different groups ($n = 5$); **i** Masson's Trichrome staining-based quantification of fibrotic tissue percentage from different groups ($n = 5$); **j** Representative images of wheat germ agglutinin (WGA) staining to visualize cardiomyocytes in sections from remote area ($n = 5$), scale bar: 10 μm. Data are presented as mean ± SD with at least triplicates. Statistical analysis was conducted by two-tailed Student's $t$-test, where *$p < 0.05$; **$p < 0.01$; ***$p < 0.001$ and ****$p < 0.0001$. **a** Created with BioRender.com. Source data are provided as a Source Data file.

membrane penetration concurrently[56]. Nevertheless, the limitations including the cytotoxic property and nonspecific membrane disruption are major barriers impeding its further development[56]. Here, the incorporation of EEPG into GluCARDIA nanosystem optimized the features of NPs by endowing inherent targeting affinity, concomitantly, grafting biocompatible properties for potential clinical translation. Compared to previous work, where the peptides were covalently conjugated on the backbone of polymers for drug delivery applications[57], the designed GluCARDIA NPs allow for one-step fabrication without further modification, which is beneficial for reducing batch-to-batch variation and facilitating green synthesis scheme. Therefore, these results suggest an existing synergistic effect from a proper combination of building blocks to fabricate hybrid NPs.

Besides of experimental screen to identify potential NPs formulation, the question still remains what patterns can be inferred from screening process and what is the underlying mechanism beneath the decision-making of each step. Despite previous studies potentially proposed several quantitative structure-property relationship (QSPR) models for different gene delivery carriers, these predictions were often refined to specific types of materials, such as lipids or polymers, and the transferability of these models to other types of materials remains illusive[58–60]. Additionally, these models typically focus on a particular function of the material, such as gene silencing efficiency, with relative less efforts focusing on a holistic evaluation for different aspects[58,61]. Here, we adopted an integrated approach, combining MD simulation and multivariate regression, to potentially propose key physicochemical descriptors from a wide type of molecules for optimizing specific properties at each screening step. Considering the limited data size, we decided to avoid complex models which are more likely to overfit, but rather adopting the simplest model[62]. This simplicity, on the other hand, may also lead to straightforward interpretations[63]. Nevertheless, further in-depth exploration of a sufficient number of formulations would be helpful to establish a precise prediction algorithm to facilitate the development of innovative NPs.

A flexible system capable of delivering siRNA therapeutics to injured tissue in an effective and selective manner has immense research and translational significance. To test the applicability of GluCARDIA NPs for potential therapeutic interventions by modulating specific biological responses in Dectin-1$^+$ macrophages[40], we adopted a murine myocardial IR model, which is one of the most common clinical scenarios and also featured with aberrant Dectin-1 macrophages activation and infiltration in injured myocardium[40]. In regarding to cardiac accumulation, GluCARDIA NPs outperformed α-glucan based nanosystem (ICG-Dextran NPs), confirming feasibility of the currently proposed method in promoting the cardiac NPs accumulation after IR. As a proof of concept, we adopted GluCARDIA NPs to encapsulate siIRF3 for genetic interrupting IRF3-dependent signaling pathway, which was abnormally activated in the setting of cardiac ischemic injury[50]. We further demonstrated that GluCARDIA-mediated genetic intervention during initial phase post-IR improved myocardial remodeling, potentially indicating the feasibility of GluCARDIA NPs as a platform for modulating cardiac reperfusion diseases.

In conclusion, we constructed a diversity-oriented combinatory formulation screening for targeted cardiac RNAi therapeutics with polysaccharide framework. Featured with automated screening process and a modular, a step-wise screening protocol, we anticipate the proposed methodology developed herein to be transferable to a wider scenario in designing polysaccharide-based NPs for different biological applications, and the generated knowledge and developed GluCARDIA NPs thereof function as a powerful tool-box for pioneering precision cardiac therapy.

## Methods
### Materials and reagents
The endosomolytic polymer EEPG was synthesized as previously reported[6], β-1,3-1,4-glucan (Barley; Low Viscosity) was purchased from Megazyme, other reversible addition fragmentation chain transfer (RAFT) polymerization agents were purchased from Sigma Aldrich. Chitosan (low molecular weight (Mw), high Mw, and Mw = 15 k), PEI (Mw = 800 and Mw = 25 k), spermine, and deferoxamine were purchased from Sigma Aldrich. Dextran-spermine and Dextran-diaminobutane were obtained as described before[13,64]. Lipid 2, Lipid 6, Lipid 8, Lipid 10, and Lipid 14 were kindly gifts from Prof. Dan Peer's lab and reported in previous study[65,66]. Cell-penetrating peptides (K9, KALA, Transportan and Penetratin) are bought from Apeptide Co., Ltd. GAPDH siRNA, scrambled siRNA, Cy5-labelled siRNA, and IRF3 siRNA were purchased from GenePharma. The sequences were used as follow: siGAPDH, sense, 5′- UGACCUCAACUACAUGGUUTT-3′; antisense, 5′-AACCAUGUAGUUGAGGUCATT-3′. Scrambled siRNA (siNC), sense, 5′- UUCUCCGAACGUGUCACGUTT-3′; antisense, 5′-ACGUGACACGUU CGGAGAATT-3′. Cy5-labelled siRNA, sense, 5′- UUCUCCGAACGUGU-CACGUTT-3′; antisense, 5′- ACGUGACACGUUCGGAGAATT-3′. IRF3 siRNA, sense, 5′-GACGCACAGAUGGCUGACUTT-3′; antisense, 5′-AGUCAGCCAUCUGUGCGUCTT-3′. The detail information of antibodies was listed in Table S8.

### Preparation of different cationic species-derived formulations
The preparation of different categories of cationic materials/EEPG paired formulations was performed by an automated microfluidics platform. Briefly, EEPG polymer was dissolved in 1 M sodium hydroxide buffer or Tris-HCL as stock solution in outer phase. Cationic materials in inner phase were dissolved by specific buffers at certain concentrations, the experimental details were showed in Table S1. The flow conditions in two phases were kept at 20 mL h⁻¹ to precisely control the assembling process of formulations fabrication. The 48-well plate as module of collection system is capable of collecting samples beneath the outlet and further transfer to the online characterization system controlled by a monitoring device.

### The device composition and workflow of robotic assisted microfluidics platform
The robotic-assisted microfluidics platform consists of a sample preparation module, a microfluidic module, and an analysis module. The detailed description of workflow is shown in Supplementary Video.

Sample preparation module will facilitate automated replenishment of outer and inner phase solutions, precise injection and control of flow rates. By employing two three-way valves, the EEPG and diverse cationic molecules can be introduced into the syringe as the outer and inner phases, respectively. This configuration enables seamless injection into microfluidic chips, facilitating subsequent sample synthesis through the convenient adjustment of valve port connection. Moreover, the precision syringe pump is programmed to seamlessly transition between preconfigured flow conditions. This well-conceived feature streamlines sample replenishment, eliminating the necessity to disassemble the syringe, which guarantees a continuous and precise injection of solutions for sample synthesis.

The microfluidic module was employed for the preparation and collection of nanoparticles (NPs). The fabrication of EEPG-based NPs was achieved by using a co-flow flow-focusing microfluidic chip as shown in Supplementary Fig. 16. In addition to sample synthesis, automated sample collection was accomplished using a 48-well plate capable of precise positioning along the horizontal plane. The holes in the plate were aligned with the point of sample discharge, facilitating precise and automatic sample collection. Undesired waste liquid was collected using a vertically positioned and suspended effluents collecting box.

The analysis module integrates an automated online characterization system that combines a robotic arm, two peristaltic pumps, and an optimized dynamic light scattering (DLS) analysis system. The robotic arm ensures precise sampling by maneuvering to various positions on the well plate and collaborates with the peristaltic pump to extract samples from the 48-well plate into the detection cell. Upon completion of the analysis, another peristaltic pump is employed to transfer the solution from the detection cell to a waste liquid cylinder. Moreover, the robotic arm and peristaltic pump can be utilized to collect and dispense deionized water from a diluent storage container. This enables the cleaning of the sample cell or the dilution of the solution under examination.

These modules were integrated and utilized in tandem to achieve an automated and continuous synthesis process, as well as screening of NPs formulations. The inner phase reservoir is designed to accommodate a total of 10 reagent tubes, with five of them containing different inner phase solutions, while the remaining five tubes are allocated for holding the cleaning solution. The synthesis procedure for the first sample at different flow rates initiates after the initial injection of the inner phase solution into the syringe through the three-way valve. To maintain the integrity and purity of the collected samples, the collection command was executed following the completion of the synthesis process. This enables the injection of residual samples in the pipeline into the well plate, while the synthetic solution at the front end is collected in the effluents collecting box. To avoid cross-contamination between samples associated with different cationic molecules, the cleaning solution from the reagent tube is drawn into the system, and the entire tube is cleaned after completing the sample synthesis for each reagent tube. A total of 7 distinct flow rate conditions are set for each cation molecule, allowing for the performance of 5 different inner phase solution syntheses per batch. This leads to the acquisition of samples and the corresponding characterization data from $5 \times 7$ wells in the 48-well plates. Following that, the solution and well plates of the inner phase reservoir are replaced, and the next batch of synthesis is carried out for the subsequent set of samples.

## Fabrication of microfluidic chip

The initial step involved designing and modeling the chip's structure using SolidWorks (Dassault Systems S.A). The fabrication process employed a projection micro-stereolithography-based 3D printing method provided by BMF Precision Tech Inc., achieving a high resolution of 10 µm. Upon completion of the printing, the chip underwent a rinsing process with isopropyl alcohol (IPA) to eliminate any residual impurities. Subsequently, the chip was cured using a UV curing device to ensure the stability and integrity of the printed structure. Finally, nitrogen gas was used to eliminate any remaining traces of IPA, ensuring the chip was free from contaminants before further application in the synthesis of EEPG-based NPs.

## Nanoparticles characterization

The hydrodynamic diameter was detected by using a Nanolink S901(Linkoptik Instruments Co., Ltd.), equipped with automatic injection, cleaning, and detection modules. Zeta potential measurements were conducted by using a Zetasizer Nano ZS (Malvern). The morphology of nanoparticles was observed by transmission electron microscopy (TEM), a carbon-coated copper TEM grid (Ted Pella) was used for the preparation of nanoparticles. The TEM analysis was conducted by the FEI Tecnai G2 F20 X-TWIN Transmission Electron Microscope (USA).

## Cell culture

Human monocyte THP-1 cell line (Cat# TIB-202, ATCC) was cultured in RPMI 1640 Medium supplemented with 10% fetal bovine serum and 1% penicillin-streptomycin (complete RPMI 1640 medium). Murine RAW 264.7 cell line (Cat# TIB-71, ATCC) was cultured in high glucose Dulbecco's Modified Eagle's Medium with GlutaMAX supplement (ThermoFisher), containing 10% fetal bovine serum and 1% penicillin-streptomycin. Murine bone marrow-derived monocytes were isolated from the tibia of C57BL/6 mice and differentiated into M0 macrophages in complete RPMI 1640 medium supplemented with 20 ng mL$^{-1}$ macrophage colony-stimulating factor (M-CSF). Bone marrow-derived macrophages (BMDMs) will be collected at day 6 after stimulation. All the cells were cultured in a humidified incubator with 5% $CO_2$ at 37 °C.

## Cytotoxicity study

For cytotoxicity study, cells were pre-seed in 96-well plates at $1 \times 10^4$ cells per well. Prior to this, human monocytes THP-1 cells were incubated with complete RPMI-1640 containing 100 ng mL$^{-1}$ of phorbol 12-myristate 13-acetate (PMA) for 24 h to differentiate into M0 macrophages. The cell viability of different cells was detected by cell counting kit-8 (CCK-8) at 48 h after incubating with representative nanoformulations, the CCK-8 assay was performed according to the manufacture's protocol.

## siRNA encapsulation and transfection

The encapsulation efficiency of siRNA loaded in different nanoparticles was quantified by Ribogreen assay as described elsewhere. For in vitro transfections, $3 \times 10^5$ cells were pre-seeded in 6-well plates, after attachment, cells were incubated with siGAPDH/siNC-encapsulated nanoparticles at a final working concentration of 50 nM siRNA. The whole cell lysates were collected at 48 h post-transfection and processed for further analysis.

## Western blot analysis

Cells were detached by scraper and lysed with cold RIPA buffer, followed by fully lysed in a FS30D bath sonicator (Thermo Fisher, USA). The total proteins were extracted at 12000 rpm, 4 °C for 5 min and quantified by using BCA Protein Assay Kit (Thermo Fisher, USA). For western blotting analysis, denatured proteins were separated in a 4–12% SDS-polyacrylamide gel electrophoresis, followed by transferring onto a polyvinylidene fluoride (PVDF) membrane, which was pre-activated by methanol. After blocking with EveryBlot Blocking buffer (Bio-Rad) for 5 min, the membranes were further incubated with diluted primary antibodies overnight at 4 °C. At the second day, after washing with $1 \times$ TBST for three times, the membranes were further incubated with corresponding secondary antibodies for 1 h at room temperature. The immunoblots signals were detected by Clarity™ Western ECL Substrate (Bio-Rad) and analyzed by ImageJ software.

## Computational aided interpretation

Chemical structures for all compounds were prepared by ChemDraw 21.0 (PerkinElmer), and their corresponding simplified molecular-input line-entry system (SMILES) representations were summarized in Supplementary Data 2. Chemical structure of each compound was energy minimized by Open Babel 2.4.0, and the MolDes of each compound were calculated by alvaDesc 2.0, and variable reduction was conducted based on the following scenarios: constant or near constant values, standard deviation less than 0.0001; at least one missing values in one of the molecules; paired correlation larger or equal to 0.95. The calculated MolDes were correlated to binary or continuous observations from experiments as discussed in the manuscript.

For molecular dynamics simulation, one EEPG molecule and one pairing cationic molecule were placed in a $6 \times 6 \times 6$ nm water box with periodic boundary conditions containing ~6800 TIP3P model water molecules and sodium or chloride counter-ions to neutralize the system. To run the simulations, the GROMACS 5.1.1 simulation package was used with the Amber99sb force field. Electrostatics and van der Waals interactions were evaluated in the default cutoff value. Simulation parameters for all the compounds were obtained from SobTop (Version [1.0 (Dev)] Available online: http://Sobereva.com/Soft). The EEPG-cationic compound configurations were energy minimized and annealed from 0 to 370 K at a constant pressure with an annealing rate of 37 K ps$^{-1}$. The temperature was kept constant for extra 50 ps, followed by decreasing the temperature to 0 K at a constant pressure with an annealing rate of 37 K ps$^{-1}$. The periodic annealing was conducted for 4 times and the temperature of the final system was further increased to 298 K at a constant pressure with an annealing rate of 29.8 K ps$^{-1}$, sequentially run for 1 ns under NPT condition. Exchange between adjacent temperature replicas was attempted every 1 ps. The time step of the simulation was 2 fs. The trajectories were saved every 1 ps, yielding a total of 1000 snapshots for production analysis. Structures were visualized in VMD.

## Animals and murine model establishment

All animal experiments were reviewed and approved by the Institutional Animal Care and Use Committee (IACUC) of Hangzhou Medical College (Animal License No. SYXK (Zhe) 2019-0011, Approval No. ZUCLA-IACUC-20010180). The animals were treated in accordance with relevant institutional and national guidelines and regulations.

Eight to ten-week-old C57BL/6 mice were purchased from The Jackson Laboratory and randomly allocated to different groups, all animal experiments were performed in compliance with the guidelines from the Institutional Animal Care and Use Committee at School of Hangzhou Medical College. All mice were housed in a pathogen-free animal facility with 40–60% humidity at a constant temperature ($22 \pm 2\,°C$), and a 12 h light/dark cycle, supplied with standard lab chow and water ad libitum. For the establishment of murine model with myocardial ischemic reperfusion injury (IR), mice were anesthetized with 1.0% to 1.5% of isoflurane gas by mechanical ventilation via a rodent respirator. After performing left thoracotomy, the left anterior descending (LAD) coronary artery visualized by a microscope was ligated with 8-0 silk suture, followed by confirming the presence of myocardial blanching. Mice were then undergoing 50 min-long LAD artery ischemia followed by reperfusion.

## Biodistribution study

The organ/subcellular distribution of GluCARDIA NPs was performed on LAD ligation induced IR mice. Briefly, C57BL/6 mice underwent IR were intravenously (i.v.) injected of 0.8 mg mL-1 ICG-Dextran NPs or ICG-GluCARDIA NPs at 24 h post-reperfusion. 8 h post i.v. administration, the mice were euthanatized and the major organs were harvested for real-time fluorescence imaging. The fluorescent signal from each organ was measured with IVIS® Spectrum CT (PerkinElmer, USA) as described elsewhere[67]. For confocal immunofluorescent imaging, 8 h post i.v. administration, mice were euthanatized and hearts were harvest for histologic section. 4′,6-diamidino-2-phenylindole (DAPI) was used to stain the nucleus, F4/80 was used to stain the macrophages and Ly-6G was used to stain the neutrophils, the detailed information of corresponding antibodies was listed in Table S8. The confocal images were taken with a Zeiss LSM 780 (Carl Zeiss, Germany) inverted confocal microscope.

## In vivo flow cytometry analysis

The subcellular uptake of nanoparticles was evaluated by flow cytometry. Briefly, GluCARDIA NPs and Dextran NPs were labelled by fluorescein isothiocyanate (FITC) for fluorescent imaging purpose. Moreover, a commercialized NPs for comparison, SM-102 LNPs was synthesized as the molar ratios of each composition in the final formulation are SM-102: DSPC: Cholesterol: DMG-PEG2000 = 50: 10: 38.15: 1.5. Mice were subjected to 50 min of ischemia followed by 24 h of reperfusion, afterwards, FITC-labelled GluCARDIA NPs/Dextran NPs/SM-102 LNPs was i.v. injected for further analysis. 8 h post-injection, organs including heart and liver were harvested for flow cytometry analysis. Single cell suspensions were prepared as described before[68]. After stained with antibodies for labeling different types of immune cells, data was acquired on a BD LSRFortessa™ X-20 System and analyzed using FlowJo software. The corresponding information of antibodies was provided in Table S8.

## In vivo treatment with GluCARDIA NPs

Scrambled siRNA (siNC)/siIRF3-loaded GluCARDIA NPs were i.v. injected to mice at 10 min, day 2, day 4 post-IR injury (siRNA dosage 2.64 μg, 0.1 mg/kg for each injection). At the 1 week and 5 weeks post-IR, the echocardiography imaging analysis was performed to monitor the cardiac function in different groups after i.v. administration. Mice were anesthetized by isoflurane, followed by gently fixing on the echo pad in a supine position. M-mode scanning was conducted as a continuous tracing to show the motion of the myocardial walls as they contract during systole and relax during diastole. The major parameters obtained from M-mode imaging include left eft ventricular end systolic volume (LVESV), left ventricular end diastolic volume (LVEDV) were then used to calculate ejection fraction (EF) to reflect cardiac contractility. The mathematical formula for calculating EF:

$$EF = [(LVEDV - LVESV)/LVEDV] \times 100$$

For immunohistological analysis, dissected heart and liver tissues were fixed in 4% paraformaldehyde and embedded in paraffin, followed by processing for tissue-section staining. The detail information of antibodies was showed in Table S8.

## RNA-sequencing transcriptome analysis

The total cell RNA isolated from heart tissues were extracted using TRLzol Reagent according to the manufacturer's protocol. DNaseI was used to eliminate DNA, followed by analyzed with Agilent 2100 Bioanalyzer to evaluate the purity and integrity of RNA samples. cDNA libraries were prepared by using the NEBNext Ultra Directional RNA Library PrepKit, the NEBNext Poly (A) mRNA Magnetic Isolation Module and NEBNextMultiplex Oligos, which were further tested by Agilent 2100 Bioanalyzer. The final cDNA libraries were established after purification and quantification in ABI StepOnePlus RealTime PCR System. The high-throughput RNA-sequencing was conducted on the Illumina HiSeq.

## Statistics and reproducibility

Two-tailed Student's unpaired $t$-test or One-way analysis of variance (AVOVA) followed by Dunnett's multiple comparison test was conducted to compare between two groups or among different groups.

The statistical analysis was performed by using GraphPad Prism (version 9.4.1). Data was presented as mean ± SD, $p < 0.05$ was considered as statistical significance. Independent experiments were conducted with a minimum of triplication each condition for comparison. The representative dataset is indicated in the Figures.

## Reporting summary

Further information on research design is available in the Nature Portfolio Reporting Summary linked to this article.

## Data availability

The authors declare that all data supporting the findings of this study are available within the paper and its supplementary information files. Source data are provided with this paper.

## Code availability

Source codes for statistical analysis and molecular dynamics simulation in Fig. 4 and Supplementary Fig 5 are available at: https://github.com/santoslabUMCG/Source-Code.

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

## Acknowledgements

H.G. acknowledges financial support from the Chinese Scholarship Council (No. 202006090004). S.L. acknowledges financial support from Seed Funding from Second Affiliated Hospital of Zhejiang University School of Medicine. Z.L. acknowledges financial support from Academy of Finland (No. 340129). M.M. acknowledges financial support from National Natural Science Foundation of China (No. 52072392) and Youth Innovation Promotion Association CAS (No. 2020255). H.C. acknowledges financial support from Shanghai International Cooperation Project (No. 23490712900). Prof. H.A.S. acknowledges financial support from Research Council of Finland (Grant No. 331151) and the UMCG Research Funds. Prof. D.P. acknowledges the support from the European Research Council (ERC Adv. Grant # 101055029), The EXPERT project (European Union's Horizon 2020 research and innovation programme (under grant agreement # 825828), ISF grant (2012/20) and the Lewis Trust for Blood Cancer. We acknowledge the valuable discussions and advice from Prof. Andre Python from the Center of Data Science at Zhejiang University regarding the analysis and interpretation of the statistical model. We acknowledge the design and presentation of Fig. 1 and Fig. 2a from BayeStat Ltd. We further thank BioRender.com for providing a platform to create the schematics used in figures.

## Author contributions

Conceptualization: H.G., Z.H.L., H.A.S.; Methodology: H.G., M.M., Z.Y.L., G.S.N., H.R.C., Z.H.L.; Investigation: H.G., S.L., D.Pan, Z.H.L.; Visualization: D.P., Z.H.L, H.A.S.; Supervision: D.P. M.M., Z.H.L., H.A.S.; Writing—original draft: H.G., Z.H.L.; Writing—review & editing: H.G., Z.H.L., H.A.S.; Funding acquisition: M.M., H.R.C., Z.H.L., D.P., H.A.S.

## Competing interests

D.P. receives licensing fees (to patents on which he was an inventor), has invested, has consulted, has been on the scientific advisory boards or boards of directors, given paid lectures or conducted research at Tel Aviv University sponsored by ART Biosciences, BioNTech SE, Earli, Kernal Biologics, Geneditor Biologics, Merck, NeoVac, Newphase, Roche,

SirTLabs Corporation and Teva Pharmaceuticals. All other authors declare no competing interests.

## Additional information

[1]Department of Biomaterials and Biomedical Technology, University Medical Center Groningen (UMCG), The Personalized Medicine Research Institute (PRECISION), University of Groningen, Ant. Deusinglaan 1, Groningen 9713 AV, The Netherlands. [2]Drug Research Program, Division of Pharmaceutical Chemistry and Technology, Faculty of Pharmacy, University of Helsinki, Helsinki FI-00014, Finland. [3]Department of Vascular Surgery, The Second Affiliated Hospital, Zhejiang University School of Medicine, Hangzhou 310009, China. [4]Shanghai Institute of Ceramics, Chinese Academy of Sciences, Shanghai 200050, China. [5]Key Laboratory of Environmental Medicine and Engineering of Ministry of Education, and Department of Nutrition and Food Hygiene, School of Public Health, Southeast University, Nanjing 210009, China. [6]Laboratory of Precision Nanomedicine, Shmunis School of Biomedicine and Cancer Research, George S. Wise Faculty of Life Sciences, Tel Aviv University, Tel Aviv 69978, Israel. [7]Department of Materials Sciences and Engineering, Iby and Aladar Fleischman Faculty of Engineering, Tel Aviv University, Tel Aviv 69978, Israel. [8]Center for Nanoscience and Nanotechnology, Tel Aviv University, Tel Aviv 69978, Israel. [9]Cancer Biology Research Center, Tel Aviv University, Tel Aviv 69978, Israel. [10]Department of Orthopedic Surgery, The Second Affiliated Hospital, Zhejiang University School of Medicine, Hangzhou 310009, China. [11]These authors contributed equally: Han Gao, Sen Li. ✉e-mail: mma@mail.sic.ac.cn; zehua.liu@helsinki.fi; h.a.santos@umcg.nl

