## [Peer Review File · Nature Communications]

Reviewers' comments:

Reviewer #1 (Remarks to the Author):

In this manuscript, Gao, Li and co-workers reported the design and optimization of polysaccharide framework-based nanoparticles for delivering RNAi therapeutics in a myocardia IR model. The authors used a robotic assisted step-wise method to optimize the NP formulation detail and demonstrate some rule of the decision-making formulation. Their candidate can target macrophage in vitro and in vivo. Finally, they delivered siRF3 to myocardia for therapeutic application, and used RNA-seq to demonstrate TGF- β related signaling after the treatment. Even this study provides some interesting method and application based on their delivery platform, however, there are many major issues should be addressed before reconsideration and publication in Nature Communications.

1. Introduction part. When the authors mentioned non-liver delivery, their expression is too overclaimed. There are still many methods/papers demonstrate that targeting of non-liver with active targeting or only endogenous targeting. Not like what they claimed about the surface modification. Some important literatures should be cited based on this problem. Cheng, Nature Nano 2020, 15, 313; Xue, JACS 2022, 144, 9926; Qiu, PNAS, 2022, 119, e2116271119.
2. It is very confused about the terminology of diversity orientation. From their experiments, it seems like they only mixed different chemical stuff together, not a concept advancement. I am not sure if it can be called "diversity-oriented", should be re-wording.
3. The novelty of the platform is limited. Peptide has already been published in their previous study (ref.3). The targeting method has already been used in their previous study (ref.5).
4. Robot assisted microfluidics platform is interesting. The authors should provide a video on how robotic assisted platform for screening to let the reader understand how it works, from the formulation to DLS analysis at least.
5. Computational assisted part is so confused. I have no idea why the authors need to do this experiment and what is the conclusion after computational analysis, and if there is any rules can be used for optimizing the formulation process. The authors need to claim it clear.
6. Figure 5d is not in good quality. Seems like lysotracker in 2h and 4h samples are too low when compared with 1h sample. The authors need to provide high resolution imaging here.
7. Please comment on the logic about why choosing 4.8 mg/kg dosage to demonstrate the biosafety.
8. The authors need to provide more flow cytometry results on what kinds of cell types uptake their particles in the heart, and liver as well (liver showed the strongest fluorescence signal) of ICG-Dextran NPs and ICG-GLUCARDIA NPs.

9. What is the key gene related to injured myocardium? And how these genes changed before and after siIRF3 NPs treatment from RNA-seq? As they used siIRF3, the authors need to provide evidence about how the IRF3 protein level changed before and after the treatment. The original RNA-seq data should also be mentioned in the supporting information to let people check if they are interested about their model from RNA-seq.

10. Another similar question about comment#8. As liver should be the most distribution of their NPs, I am wondering if siIRF3 NP delivery influence some function of liver as the population of macrophage is also obvious. How does it influence the IRF3 level in the liver? Does it change some signaling pathway in the liver after siIRF3 NP delivery?

11. The authors need to use some benchmark NP for better comparison on siIRF3 delivery to the injured myocardial.

Reviewer #2 (Remarks to the Author):

This work introduces an innovative ionotropic gelation approach for targeted nanoparticles in precision cardiac therapy. The combinatorial formulation screen identifies potent gene delivery cargo, with GluCARDIA NPs showing promise in ameliorating cardiac reperfusion damages. The work offers a valuable formulation design strategy for polysaccharides. However, from the point of view of the high-throughput (HTP) screening platform, the reviewer cannot evaluate it sufficiently because the information

for the HTP platform is lacking. In addition, the overall structure of the HTP system is composed of commercial devices such as pumps, valves, robotic arms, and even the well-known flow-focusing microfluidic chip, so the reviewer cannot find out any novelty and technical advancement in their system. This manuscript cannot give any insight and critical contribution to the HTP microfluidic society.

In terms of the applications of the resultant combinatorial NPs for the targeted cardiac RNAi therapeutics, their research was performed systematically, and some results are interesting.

Here are some issues to be addressed to strengthen the manuscript.

1. Even though the authors put some references, it is more helpful to provide additional information about the microfluidics chip, including details such as size, depth, and other relevant parameters (Fig. 2a), and chip fabrication in the Supplementary Materials.

2. In Fig. 2a, the high-throughput screening system is depicted with several inner-phase solutions in different reservoirs. However, it appears that all reactant reservoirs share a common tubing connected to the syringe pump. This design may lead to cross-contamination if some amount of

reactant remains in the tubing. Could you provide clarification or discuss any measures taken to address and minimize the potential for cross-contamination in the experimental setup?

3. In the manuscript, a section titled "A robotic-assisted microfluidics platform for nanoparticle formation screening" is highlighted as a key result of the experiment. However, the role of the robotic arm appears to be insufficiently described and emphasized. The authors should provide clarification on the specific functions and contributions of the robotic arm within the context of the manuscript.

Minor issues are:

4. Figure legend for Fig. 3e is missing.

5. What is the meaning of KD in Fig. 4a?

6. The author should furnish more detailed information about the siRNA sequences employed in the experiments.

Reviewer #3 (Remarks to the Author):

In their paper entitled "Diversity-oriented combinatorial formulation screen for cardiac RNAi therapeutics with polysaccharide framework", Gao et al use a computational approach to generate a series of nanoparticles capable of delivering siRNA to the myocardium following injury. The authors target Dectin-1 for delivery to post-infarcted macrophages and show in vivo knockdown of a pro-inflammatory gene and improvements in cardiac function. The screening process is interesting, yet there are many unresolved questions that dampen enthusiasm.

1. If the authors knew a priori that they wanted to target Dectin-1, why the need for a long screening process rather than working to modify an already existing NP to target this protein?

2. The in vitro work is convincing that nanoparticles are created with siRNA, but testing on just 2 cell types is not convincing. Additionally, if the eventual goal is human translation, the negative effects on THP-1 are not encouraging. Could the authors discuss why they are creating a system that seemingly targets mouse macrophages?

3. The in vivo delivery needs much more quantification. A single time-point measured semi-quantitatively is not encouraging. The authors should measure %id/g in each organ as right now the heart seems to have taken up very little of the total NP population.

4. There is no timecourse of delivery. The authors delivered 24 hours after IR, but macrophages peak 3-7 days post-injury in most published studies. They do not examine any PK or PD of the NP.

5. There are no in vivo macrophage results or examination of infarct size/cell death. The authors have created a macrophage-targeted material, but do not look at macrophage polarization in vivo or any inflammatory markers. Further, macrophages play a big role in acute cardiomyocyte cell death but this is not examined.

6. If the authors created a better targeting system, they should compare to an existing system as a control. There are countless papers of NP targeting to the heart that the authors could use a control. The premise is that they have done something much better through their approach (which they may have) but do not have any comparisons.

RESPONSE TO THE REVIEWER'S COMMENTS

Reviewer #1:

In this manuscript, Gao, Li and co-workers reported the design and optimization of polysaccharide framework-based nanoparticles for delivering RNAi therapeutics in a myocardia IR model. The authors used a robotic assisted step-wise method to optimize the NP formulation detail and demonstrate some rule of the decision-making formulation. Their candidate can target macrophage in vitro and in vivo. Finally, they delivered siIRF3 to myocardia for therapeutic application, and used RNA-seq to demonstrate TGF- β related signaling after the treatment. Even this study provides some interesting method and application based on their delivery platform, however, there are many major issues should be addressed before reconsideration and publication in Nature Communications.

Answer: Thank you for your time and efforts in reviewing the manuscript. The insightful commentaries will undoubtedly help us to improve the overall quality of the manuscript. Specific discussion of your concerns and suggestions are present in point-by-point response below.

1. Introduction part. When the authors mentioned non-liver delivery, their expression is too overclaimed. There are still many methods/papers demonstrate that targeting of non-liver with active targeting or only endogenous targeting. Not like what they claimed about the surface modification. Some important literatures should be cited based on this problem. Cheng, Nature Nano 2020, 15, 313; Xue, JACS 2022, 144, 9926; Qiu, PNAS, 2022, 119, e2116271119.

Answer: Thank you for your insightful comments and for bringing to our attention the significant studies by Cheng et al. (Nature Nanotech 2020, 15, 313), Xue et al. (JACS 2022, 144, 9926), and Qiu et al. (PNAS, 2022, 119, e2116271119). We appreciate the opportunity to address your concerns regarding our statements about non-liver targeting and surface modification in nanoparticle delivery systems.

We agree with the reviewer that different studies have been evaluated for engineering selective nanoformulations with different methods including tuning the internal/external charge with different "SORT lipid" (Cheng, Nature Nano 2020, 15, 313), conjugating with targeting ligand (Xue, JACS 2022, 144, 9926) or manipulating protein corona on the surface (Qiu, PNAS, 2022, 119, e2116271119). And we also would like to propose that in the current study, we provided an alternative method by tailoring the choice and chemical structure of polysaccharides as

building blocks, leveraging the exclusive recognition of certain glycans by specific lectins, which may be abnormally expressed in lesioned sites, enabling a targeted delivery without subsequent surface modifications. More specifically, the currently proposed strategy aided the screen process and identified a nanosystem GluCARDIA simply composed by two components and fabricated in a one-step manner. This “polysaccharide-framework” principle of design may simplify the fabrication scheme and reduce batch-to-batch variation, highlighting an advantage of using polysaccharides to create non-viral NPs for targeted genetic interventions.

Once again, we are grateful for your valuable feedback, which has undoubtedly strengthened the quality of our manuscript. Upon reviewing these notable studies, we have revised our manuscript to present a more balanced view of the current landscape in gene delivery. We will also acknowledge the contributions of these studies. Specifically, we have included references to these papers in the introduction section of our manuscript (Page 3, line 7-12).

“Different strategies, including tuning the internal/external charge with different lipids to achieve specific organ tropism, targeting ligand modification or manipulating protein corona on the surface, have been adopted to mediate target cell entry in desired site. Of note, as an alternative method, previous studies proposed a simplified NPs design principle by tailoring the choice and chemical structure of polysaccharides as building blocks to facilitate the targeted delivery.”.

The corresponding information has been added in the revised manuscript and labelled with change track. We hope that these changes satisfactorily address your concerns.

2. It is very confused about the terminology of diversity orientation. From their experiments, it seems like they only mixed different chemical stuff together, not a concept advancement. I am not sure if it can be called “diversity-oriented”, should be re-wording.

Answer: Thank you for your comment regarding the use of the term "diversity-oriented" in our manuscript. We appreciate the opportunity to clarify the concept and its application in our study.

The term “diversity-oriented” in our research refers to the strategy of exploring a wide range of ionotropic pairings and directly conduct a parallel comparison to find the optimum formulation and combination. While traditional approaches have often focused on a limited set of specific polysaccharide-counterion interactions, our method aims to expand this scope in a broader scenario. We employ a combinatorial approach that encompasses a diverse collection

of molecules with varied chemical structures ranging from small molecules, lipids, polymers and peptides. Our use of the term "diversity-oriented" aims to capture this shift from a narrow, specific focus to a broader, more inclusive approach. It reflects our methodology of integrating a wide array of chemical entities to identify the most effective combinations for siRNA carriers. This approach is not merely a mixture of different chemicals but a systematic exploration of diverse ionotropic interactions to uncover novel and efficient siRNA delivery systems.

Considering the reviewer's feedback, we have also added further explanation to emphasize the conceptual advancement this approach represents in the field of siRNA carrier development and labelled with changed track (Page 3, line 19-27).

“However, to date, much of these works have centered on formulation optimization in specific ionotropic pairings, which lacks comparative analyses between different ionotropic pairings for more efficient formulation optimization. In contrast, a “diversity-oriented” combinatorial optimization approach, which involves screening a distinct collection of molecules characterized by a wide spectrum of chemical structures, may expand the range of potential ionotropic interactions and facilitate a parallel comparison between different pairings. By diversifying the investigated chemical interactions, this methodology enhances the probability of uncovering unique and optimum candidates for siRNA carrier development.”.

Meanwhile, we still agree with the reviewer regarding to the clearness of the term, upon such consideration, we modify the title of the manuscript into “Comparative optimization of polysaccharide-based nanoformulations for cardiac RNAi therapy.”

We hope this clarification addresses your concerns and accurately conveys the innovative aspect of our research approach.

3. The novelty of the platform is limited. Peptide has already been published in their previous study (ref.3). The targeting method has already been used in their previous study (ref.5).
--

Answer: Thank you for your comments regarding the perceived novelty of our platform in relation to our previous work. First, we would like to clarify that in our previous publication (ref.3, Journal of Controlled Release 357 (2023) 120–132), the peptide KALA was not utilized. Moreover, we would like to highlight that the central innovation of our current study is not merely in introducing a new targeting ligand or designing a specific gene delivery system. Instead, the focus lies in employing a fully automated microfluidic platform to screen and

validate effective gene delivery vehicles through a non-biased approach. This methodology enabled us to systematically evaluate over 161 formulations, leading to the identification of an optimal formulation with corresponding screening process elaboration. The newly identified GluCARDIA NPs, a result of this expansive screening, demonstrated clear advantages comparing to conventional formulation optimization process by achieving superior gene silencing efficacy with a 33% reduction in siRNA dosage and 70% less NPs mass (Journal of Controlled Release 357 (2023) 120–132). This underscores the efficacy of a diversity-oriented screening approach in identifying optimal conditions in a more efficient and robust manner. Our findings contribute a practical framework for the development of polysaccharide-based nanosystems for targeted RNA interference. The methodology and the resultant formulation not only enhance the efficiency and control of the delivery process but also pave the way for the development of novel active targeting NPs focused on myocardial repair and protection. We believe these aspects significantly underscore the novel contributions of our current study, distinct from our previous works, and provide meaningful advancements in the field of nanoparticle-based gene delivery.

We hope this response adequately addresses the concerns raised and further clarifies the novelty and significance of our research.

4. Robot assisted microfluidics platform is interesting. The authors should provide a video on how robotic assisted platform for screening to let the reader understand how it works, from the formulation to DLS analysis at least.

Answer: Thank you for your valuable comments regarding the robotic assisted microfluidics platform in our study. We acknowledge the raised concerns about the workflow of this platform and appreciate the opportunity to provide a video to clearly show how it works as an integrated platform. Considering the reviewer's suggestion, the video has been included in the resubmitted manuscript as **Supplementary Video S1**.

We hope this response adequately addresses the raised concerns and further elucidates the workflow of the robotic-assisted microfluidics platform.

5. Computational assisted part is so confused. I have no idea why the authors need to do this experiment and what is the conclusion after computational analysis, and if there is any rules can be used for optimizing the formulation process. The authors need to claim it clear.

Answer: Thank you for your valuable comments regarding the computational analysis of our study. We acknowledge the concerns raised about the clarity of this section and appreciate the opportunity to provide a more detailed explanation of our computational analysis and its significance.

In the current study, we conducted an extensive screening process, evaluating a total of 161 formulations through 4 rounds of screening. Each round was dedicated to assessing a specific aspect of the formulation, including NPs formation capability, siRNA encapsulation efficiency, biocompatibility, and gene knockdown efficiency. During the screening process, we noticed several phenomena that were not readily explicable with conventional mechanistic interpretations. For instance, despite ionotropic gelation is usually considered as an interaction process facilitated by two electronically opposite compounds, we observed that certain formulations, despite holding similar N/P ratios, failed to form stable NPs. Similarly, in regarding to siRNA encapsulation efficiency, we noticed significant variations among different starting materials with comparable physicochemical properties, such as molecular weight, surface charge, and chemical composition. And the complexity was further amplified by the diversity-oriented approach of our formulation screening, which included compounds with distinct molecular categories.

Upon such consideration, we turned to computational tools to try to understand the potential patterns and mechanisms in each process. However, it is important to note that the limited size of our dataset presents challenges in developing a robust algorithm for precise predicting cationic compound behavior. Therefore, our primary objective with the computational analysis was to rather to provide potential scientific understanding from the screening process rather than establishing on algorithmic prediction.

First, to help the potential readers in better visualizing the screening procedure, we have created a decision tree (DT) diagram (**Fig. 4a**) to summarizes the process. In this figure, dark blue squares represent the positive sets identified at each stage of screening, while grey squares denote the negative sets. Additionally, light blue diamonds are used to delineate the specific criteria applied at each screening juncture. And **Fig. 4b** summarized the names and identifiers of the cationic substances tested in our study.

For NPs formation, the phenomenon can be divided into 3 categories: cationic reagents that did not form stable NPs with EEPG at any concentration (Group 1: samples 15, 16, 17, and 20), and cationic compounds formed colloidal stable NPs only at N/P ratios significantly deviating

from unity (<0.1 or >10 , Group 2: samples 14, 18, and 19), and cationic reagents that can steadily form into NPs (Group 3: samples 1-13, **Fig. 1 for Review**).

Fig. 1 for Review: Categories of different samples in the screening process for determining the NPs size and formation.

Numerical values of molecular descriptors are used to quantitatively describe the physical and chemical information of the molecules. Since molecular descriptor (MolDes) is the most commonly adopted mathematical representations to quantitatively describe the physical and chemical information of the molecules (Todeschini R, Consonni V. Handbook of molecular descriptors[M]. John Wiley & Sons, 2008.), initially, we considered whether a MolDes based quantitative structure-activity relationship (QSAR) model could elucidate this phenomenon. MolDes for each compound were generated using alvaDesc, followed by high throughput independent feature selection with logistic regression facilitated by R, using binary NPs formation as the label. In total 659 MolDes was adopted for feature screening with binary NPs formation as label, however, even the MolDes with the lowest p-value (indicating the MolDes holds the most significant association with the NPs formation process) only showed a receiver operating characteristic (ROC) curve with an area under the curve (AUC, indicating the performance of the MolDes in distinguishing between the positive and negative classes, where $AUC = 0.5$ means equivalent to random guessing) of only 0.85 (**Supplementary Fig. 5a-b**), and the ambiguous actual physical and chemical significance of the top 1 MolDes (RTs+, indicates R maximal index weighted by intrinsic-state) further limits the interpretability of this model. Therefore, we seek for alternative method by employing short-term (1 ns) molecular dynamics (MD) simulations. These simulations aimed to quantify the non-covalent interaction potentials between the polyelectrolyte EEPG and various cationic compounds. The focus was on two dominant interactions: electrostatic (measured by Coulomb's potential) and van der

Waals (assessed by Lennard-Jones, LJ potential). After the simulations, we observed a distinct pattern in the interaction potentials. For the compounds that cannot form into NPs (Group 1 and Group 2), the Coulomb potential between EEPG and the counter-reagents often reached or exceeded zero. This contrasted with the values for the compounds that can form NPs with EEPG (Group 3), as depicted in **Fig. 4c**. This discrepancy indicated a minimal electrostatic contribution in the formation of stable NPs within the negative subsets. Further analyses were conducted to distinguish Group 1 and Group 2. Interestingly, in the non-electronic binding group, the inner LJ potentials for EEPG were significantly lower, suggesting a propensity for EEPG self-aggregation (**Fig. 4d**). This finding implies that the formation of NPs in these cases could be partly driven by EEPG's tendency to self-aggregate through non-polar interactions. **In conclusion, from the current results, we propose MD simulations may be adopted as a valuable supplementary tool for initial evaluations and insights into the inotropic gelation based NPs formation process, which is mainly reflected by observing the Coulomb's potential during the simulation.**

Next, we try to elucidate the critical factors in governing the siRNA encapsulation efficiency (EE%). Considering that the data generated from MD simulations may offer supplementary insights beyond the conventional MolDes computed by alvaDesc, we integrated 8 extra parameters derived from MD simulations into the overall set of MolDes, which yielded overall 667 MolDes. The set of MolDes were then utilized for independent feature selection, with encapsulation efficiency (EE%) as the variable of interest. The MD derived parameters include:

1. Average Coulomb's potential of EEPG, which refers the potential energy due to electrostatic interactions within EEPG chain;
2. Average Lennard-Jones potential of EEPG, which refers the potential energy due to van der Waals interaction, which is correlated to the distance between a pair of neutral atoms;
3. Average Coulomb's potential of between EEPG and counter-reagents, which refers the potential energy due to the intramolecular electrostatic interaction;
4. Average Lennard-Jones potential between EEPG and counter-reagents, which refers the potential energy due to the intramolecular van der Waals interaction.
5. Change of Coulomb's potential of EEPG after 1 ns simulation, which refers the potential energy alternation due to electrostatic interactions within EEPG chain;
6. Change of Lennard-Jones potential of EEPG after 1 ns simulation, which refers the potential energy alternation due to van der Waals interaction;

7. Change of Coulomb's potential of between EEPG and counter-reagents after 1 ns simulation, which refers the potential energy alternation due to the intramolecular electrostatic interaction;
8. Change Lennard-Jones potential between EEPG and counter-reagents, which refers the potential energy alteration due to the intramolecular van der Waals interaction.

Independent feature selection analysis identified that among all the 667 MolDes, the average Lennard-Jones (LJ) potential between EEPG and the counter-reagents exhibited the highest correlation coefficient ($r^2 = 0.7$, **Fig. 4e**). This value significantly surpasses those of MolDes traditionally calculated, suggesting a more robust predictor of EE%.

Considering the LJ potential is highly correlated with the distance between two molecules, we hypothesize this correlation might be attributable to steric repulsion exerted by the cationic reagents during siRNA encapsulation, where a stronger interaction between EEPG and the cationic compound could introduce spatial constraints for siRNA accommodation, adversely impacting encapsulation.

To confirm our hypothesis, we separately visualized binding of EEPG with KALA and Transportan. These two peptides were selected for comparison based on their similar molecular weight, hydrophobicity (LogP) and surface charges, yet markedly different EE% (KALA, EE% = 95%; Transportan, EE% = 25%). Notably, unlike the EEPG-KALA complex, where KALA aligns along one tail of EEPG in a flexible configuration, the EEPG-Transportan complex assumes a more compact arrangement (**Fig. 4f**). This structural variance is evidenced by the EEPG-Transportan complex establishing over tenfold more close contacts ($< 6 \text{ \AA}$) than its EEPG-KALA counterpart (**Supplementary Fig. 5c**), indicating the influence of molecular interactions on siRNA encapsulation efficiency.

In our sequential investigation into the factors contributing to the observed biocompatibility of various formulations, we employed a similar independent feature selection methodology, utilizing the IC50 values as labels and the MolDes of each cationic compound as variables. Notably, previous studies have investigated the use of MolDes to establish correlations between physicochemical properties and toxicity, and in several previous profound studies, molecular weight (Mw), lipophilicity (log P), and basicity (pKa of the strongest base) have been highlighted as significant predictors of toxicity across numerous pharmaceutical research findings (*Drug Discov Today* 17, 325-335 (2012); *Bioorg Med Chem Lett* 18, 4872-4875 (2008)). Among which, the basicity has been acknowledged as the major factor for negative regulating the biocompatibility of the materials.

However, our analysis introduces a novel insight, suggesting the critical role of the quantity and topological distribution of negatively charged atoms within a compound in determining its biocompatibility. This notion is supported by the molecular descriptor *qnmax*, which quantifies the maximum negative charge of a compound, exhibiting the highest coefficient of determination ($r^2 = 0.93$, **Fig. 4g, h, Supplementary Table 7**), which is significantly higher than *qpmax*, which quantifies the maximum positive charge of a compound; *Qpos*, which quantifies the total positive charge of a compound and *MLOGP*, which describes the hydrophobicity of the compound. Additionally, the descriptor SPP, indicative of the absolute difference between the maximum negative and positive charges, emerged as another key factor correlated with biocompatibility (**Supplementary Table 7**). Further analysis identified other significant molecular descriptors, including B02[O-O] and B10[N-N], denoting the presence or absence of O-O and N-N bonds at specific topological distances, respectively (**Supplementary Table 7**).

While the contributions of hydrophobicity and overall positive molecular charges to biocompatibility assessment are well-recognized, our findings propose that the balance and distribution of a compound's negative and positive components are equally crucial in influencing its biocompatibility.

This perspective offers a refined understanding of the molecular features that govern the biocompatibility of cationic compounds, which may further provide the visional insights in modifying the compounds for promoting its biocompatibility.

In our last pursuit to elucidate the relationship between the chemical properties of cationic counterparts and the gene knockdown efficacy of the resultant NPs, we employed knockdown efficiency of each formulation as the target and MolDes of each cationic compound as predictors. Coefficients of determination of each MolDes is adopted for evaluating the impact of each MolDes, as it represents the proportion of the variance for a dependent variable (in the current case, gene knockdown efficiency) to be affected by an independent variable (in the current case, MolDes of each compound). Independent feature selection noted that among the top five MolDes demonstrating the highest coefficients of determination, three (TDB10p, E3m, and H2p) were three-dimensional (3D) molecular descriptors, with TDB10p showing the most significant correlation, evidenced by an r^2 value of 0.93 (**Fig. 4i-j**). These descriptors involve both the 3D geometric and chemical characteristics of molecules, applying various weighting schemes, such as polarizability (for TDB10p and H2p) and atomic mass (for E3m). The comprehensive elucidation of each MolDes is provided in **Supplementary Table 7**.

This is in consistent with previous studies, indicating that the gene knockdown efficiency from biomaterials is not necessarily correlated with their chemical components. For example, previous research has evaluated the transfection efficiency of different cationic polyelectrolytes, and the results suggested even those with identical chemical compositions, their corresponding gene knockdown efficiency can be heavily altered (*Nature Biotechnology* 17, 784-787 (1999); *Biomacromolecules* 15, 1328-1336 (2014)), and the potential mechanism may lay beneath their distinct 3D orientation. Therefore, while for MolDes describing explicit chemical or physical characteristics could offer better interpretability, they may fall short in capturing the complex behaviors observed in our experiments. This is further supported by the unsatisfactory r^2 values obtained from MolDes describing hydrophobicity, polar surface area, and positive charges (**Supplementary Fig. 5d, Supplementary Table 7**).

Therefore, we propose here further studies should focus on 3D structural attributes of cationic compounds in predicting the transfection efficiency of engineered nanosystems.

This insight holds specific scientific value for constructing statistical model for materials prediction and optimization, as reducing the number of features, which is also known as feature selection or dimensionality reduction, can significantly promote the performance, interpretability, and generalization ability of a model.

6. Figure 5d is not in good quality. Seems like lysotracker in 2h and 4h samples are too low when compared with 1h sample. The authors need to provide high resolution imaging here.

Answer: Thank you for bringing up this question and for the time and effort in helping us improve the manuscript. Based on the reviewer's suggestion, we replaced the figures in Figure 5d with higher resolution, we are grateful for this valuable comment and believe this will improve the overall quality of our manuscript.

7. Please comment on the logic about why choosing 4.8 mg/kg dosage to demonstrate the biosafety.

Answer: Thank you for your comments regarding the rationale of choosing 4.8 mg/kg dosage in our study. We appreciate the opportunity to clarify the logic about the biosafety experiment. For *in vivo* administration of GluCARDIA NPs, prior to therapeutic tests, we evaluated the biosafety of NPs at the dosage of 4.8 mg kg⁻¹ was determined by the dosage of NPs necessary to encapsulate enough siRNA for achieving effective gene silencing in the following studies.

Therefore, the NPs dose was pre-determined to be 0.12 mg per mouse (corresponds to 4.8 mg kg⁻¹) for each injection. To make it clear, the relevant discussion has been added to the manuscript and labeled with changed track. We hope these changes satisfactorily address your concerns (Page 21, line 11-13).

“Prior to this, the biosafety property was evaluated with intravenous (i.v.) administration of GluCARDIA NPs at the dosage of 4.8 mg kg⁻¹ (pre-determined by the potential siRNA dosage of 2.64 µg per mouse).”

8. The authors need to provide more flow cytometry results on what kinds of cell types uptake their particles in the heart, and liver as well (liver showed the strongest fluorescence signal) of ICG-Dextran NPs and ICG-GLUCARDIA NPs.

Answer: Thank you for your insightful comments regarding the subcellular uptake of the developed nanoparticles in our study. We acknowledge the valuable comments and appreciate the opportunity to provide more detailed information on specific cellular uptake of our nanoparticles.

Considering the reviewer’s suggestion, we evaluated the subcellular uptake of different NPs after *in vivo* administration. To this end, fluorescein isothiocyanate (FITC) labelled GluCARDIA NPs was synthesized for fluorescent imaging purpose. As a comparison, we also developed FITC-labelled Dextran NPs, as a negative polysaccharide control without Dectin-1 targeting affinity, and FITC-labelled SM-102 LNPs, as a comparative benchmark for commercialized NPs in gene delivery. Mice were subjected to 50 min of ischemia followed by 24 h of reperfusion, afterwards, FITC-labelled GluCARDIA NPs/Dextran NPs/SM-102 LNPs was i.v. injected for further analysis. 8 h post-injection, organs including heart and liver were harvested for flow cytometry analysis. The gating strategy to identify different immune cell subsets by flow cytometry experiments was shown as Fig. 2a for Review.

Comparing to Dextran NPs and SM-102 LNPs, GluCARDIA NPs demonstrated a distinct cellular interaction pattern with cardiac immune cells following myocardial IR. GluCARDIA NPs mainly co-localized with monocytic cells. Meanwhile, we also observed an enhanced monocytic interaction from GluCARDIA NPs comparing to Dextran NPs and SM-102 LNPs. (Fig. 2b for Review). Notably, GluCARDIA NPs also exhibited substantially higher recognition by dendritic cells, which are also known as major Dectin-1⁺ cellular subsets upon

sterile inflammation (*Nat. Commun.* 2016; 7:12368), while exhibiting a comparable cellular uptake with other immune cells (*Circulation.* 2019; 139:663–678).

Regarding to the potential cellular interaction with hepatic immune cells, despite the major colocalization was still observed in monocytic cells, in contrary to the enhanced monocytic targeting efficiency in lesioned myocardium, GlucCARDIA NPs exhibited a similar monocytic internalization efficiency comparing to SM-102 LNPs (**Fig. 2c for Review**). This may be attributed to the restrained Dectin-1 expression profile in hepatic macrophages upon sterile injury (*J. Control. Release.* 2023; 357:120). In conclusion, here we have demonstrated that in the setting of myocardial IR, our designed GluCARDIA NPs are primarily recognized by monocytic cells, significantly exceeding its uptake in other immune cells. Moreover, GluCARDIA NPs exhibited substantially higher targeting efficiency towards cardiac monocytes comparing to α -polysaccharides (dextran) based NPs and commercialized benchmark NPs (SM-102 LNPs), offering a promise delivery system for efficient cardiac gene delivery post IR.

The corresponding information has been added in the manuscript (Page 22, line 22-34; Page 23, line 1-13) and assigned as **Supplementary Fig. 11**. All changes are labelled with change track. We hope that these changes satisfactorily address your concerns.

Fig. 2 for Review. (a) Gating strategy for flow cytometry analysis. All cells were extracted from the heart and liver, and further labeled with antibodies as described in Methods. Monocytes was identified as CD45+CD11b+Ly6G-Ly6clow to high, neutrophils were identified as CD45+CD11b+Ly6G+, T cells were identified as CD45+CD11b-CD3+, B cells were identified as CD45+CD11b-B220+, NK cells were identified as CD45+CD11b-NK1.1+, DCs were identified as CD45+CD11b-CD11c+MHC II+. Quantitative analysis of immune cell uptake of SM-102 LNPs/Dextran NPs/GluCARDIA NPs was performed on (b) heart and (c) liver. Data are presented as mean \pm SD. ns: no significance; *, $p < 0.05$, **, $p < 0.01$, ****, $p < 0.0001$.

9. What is the key gene related to injured myocardium? And how these genes changed before and after siIRF3 NPs treatment from RNA-seq? As they used siIRF3, the authors need to provide evidence about how the IRF3 protein level changed before and after the treatment. The original RNA-seq data should also be mentioned in the supporting information to let people check if they are interested about their model from RNA-seq.

Answer: Thank you for your valuable comments regarding the RNA-seq of our study. Exploring the dominant genes for mediating myocardial IR has been intensively investigated, with constant validation of signaling pathways confirmed with therapeutic potential. Previous study identified the aberrant activation of IRF3 and type I interferons as one of the hallmarks for sterile cardiac injuries, and the overt activation of IRF3 fuels a fatal response towards myocardial injuries, where the interruption of IRF3-dependent signaling (by testing in transgenic IRF3^{-/-} mice) resulted in an improved cardiac function post ischemic cardiac injury (*Nat. Med.* 2017; 23:1481). More importantly, the activation of this IRF3–interferon axis is distinctly observed in cardiac macrophages (*Nat. Med.* 2017; 23:1481), highlighting a targeted therapeutic opportunity.

Leveraging this insight, and given the macrophage-targeting capabilities of GluCARDIA NPs, our current investigation employs siIRF3 to assess the efficacy of cardiac gene delivery in mitigating the consequences of myocardial IR. This strategic approach underscores the potential of targeted gene silencing in improving the prognosis of myocardial IR injuries.

Regarding the expression level of IRF3, previous studies confirmed the quiescent expression of IRF3 in healthy heart, and the obvious activation of IRF3 was usually observed shortly after the onset of synchronous cell death in the heart (*Nat. Med.* 2017; 23:1481). In the current study, we performed western blotting analysis to compare expression levels of IRF3 in different groups. As shown in **Fig. 6i-j** in the original manuscript, after *i.v.* administration with different formulations, the protein levels of IRF3 and p-IRF3 showed marked decrease, which was not observed from neither GluCARDIA-siNC nor free siIRF3 group, confirming the cardiac IRF3 silencing efficacy from GluCARDIA NPs in the setting of myocardial IR.

Meanwhile, in anticipation of potential interest from the scientific community in our RNA-seq results, we have included the raw RNA-seq data within the supplementary materials of the revised manuscript (**Supplementary Data 1**).

We hope this clarification addresses your concerns and accurately conveys the results of RNA-seq analysis in our research.

10. Another similar question about comment#8. As liver should be the most distribution of their NPs, I am wondering if siIRF3 NP delivery influence some function of liver as the population of macrophage is also obvious. How does it influence the IRF3 level in the liver? Does it change some signaling pathway in the liver after siIRF3 NP delivery?

Answer: Thank you for your valuable comments regarding the effects of GluCARDIA-siIRF3 on liver. We acknowledge the raised concerns and appreciate the opportunity to provide more detailed information about gene expression profiling in liver.

In light of your suggestion, we further performed RNA-seq analysis to collect a genome-wide expression profiling in liver upon the administration of GluCARDIA-siIRF3. To this end, mice underwent same IR scenario were *i.v.* injected with GluCARDIA-siIRF3. 24 h post administration, liver was harvested for conducting RNA-seq analysis. As shown in **Fig. 3a for Review**, no significant difference was observed in IRF3 expression between the IR and GluCARDIA-siIRF3 group. no significant differences were observed between the two groups in the expression of upstream genes of IRF3 (Tbk1, **Fig. 3b for Review**) or downstream IRF3-regulated genes (Ifna4, Ifnar1, Jak1, **Fig. 3c-e for Review**), which may be attributed to the baseline expression profiling of type I interferons regulated genes in healthy liver (*Sci. Transl. Med.*, 2022, 14:eabh3831)).

To further elucidate the transcriptional landscape in the liver after the administration of GluCARDIA-siIRF3, we plotted the transcriptional level across a wider range of genes (**Fig. 3f for Review**). The results demonstrated that the expression of key genes involved in inflammation, including chemokines (C-C motif ligand, Ccl; Ccl2, Ccl5, Ccl12, C-X-C motif receptor, Cxcr; Cxcr2, C-X-C motif ligand; Cxcl5), pro-inflammatory cytokines (interleukin, IL; IL1b, IL6, tumor necrosis factor; TNF), fibrosis progression (transforming growth factor beta 1; Tgfb1, matrix metalloproteinases 9; MMP9, actin alpha 2; Acta2), and apoptosis related genes (caspase 3/9; Casp3/9, Bax, Apaf1) were comparable between control (CTRL) and GluCARDIA-siIRF3 groups. To interpret the potential subtle changes upon GluCARDIA-siIRF3 administration, Gene Set Enrichment Analysis (GSEA) was adopted to concordant differences between CTRL and GluCARDIA-siIRF3 groups. The results suggested the priori defined set of genes including pro-inflammation (leukocytes transendothelial migration),

fibrosis (TGF-beta signaling pathway), apoptosis and key liver function (glycerolipid metabolism) were not affected by GluCARDIA-siIRF3 administration (**Fig. 3g for Review**).

The corresponding information has been added in the manuscript (Page 22, line 10-13). We hope this clarification addresses your concerns and accurately conveys the impact of liver RNA-seq analysis in our research.

Meanwhile, in anticipation of potential interest from the scientific community in our RNA-seq results, we have included the raw RNA-seq data within the supplementary materials of the revised manuscript (**Supplementary Data 3**).

Fig. 3 for Review. (a-e) Expression level of Irf3 and Irf3 related genes including Tbk1, Ifnar1, Ifna4 and Jak between IR and GluCARDIA-siIRF3 group; (f) Heat map of chemokines, cytokines, fibrosis progression and apoptosis-related gene expressions. (g) GSEA for indicated

Gene Ontology (GO) or Kyoto Encyclopedia of Genes and Genomes (KEGG) defined gene clusters.

11. The authors need to use some benchmark NP for better comparison on siIRF3 delivery to the injured myocardial.

Answer: Thank you for your insightful comments regarding the use of benchmark NPs for comparison on targeted delivery to the injured myocardial.

As discussed previously, we evaluated the potential interaction between different NPs and cardiac immune cells in lesioned myocardium, and as a representative benchmark NPs used for gene delivery, SM-102 LNPs, which are applied in mRNA-based COVID-19 vaccines, such as the Pfizer-BioNTech and Moderna vaccines, has been adopted in the current research and compared with GluCARDIA NPs in regarding to cardiac targeting efficiency. Comparing to SM-102 LNPs, GluCARDIA NPs exhibited superior interaction with monocytic cells and dendritic cells, which are known as Dectin-1⁺ cells, whereas this phenomenon was not observed in other immune cells, indicating the Dectin-1 targeting capability from GluCARDIA NPs and the concomitantly induced higher cardiac accumulation comparing to benchmark LNPs, which may fulfill our envisioned design and is critical for targeted gene delivery (**Fig. 2b for Review**).

The corresponding information has been added in the manuscript (Page 22, line 22-34; Page 23, line 1-2) and labelled with change track. We hope that these changes satisfactorily address your concerns.

Fig. 2b for Review. Quantitative analysis of immune cell uptake of SM-102 LNPs/Dextran NPs/GluCARDIA NPs was performed on heart. Data are presented as mean \pm SD. ns: no significance; *, $p < 0.05$, **, $p < 0.01$, ****, $p < 0.0001$.

RESPONSE TO THE REVIEWER'S COMMENTS

Reviewer #2:

This work introduces an innovative ionotropic gelation approach for targeted nanoparticles in precision cardiac therapy. The combinatorial formulation screen identifies potent gene delivery cargo, with GluCARDIA NPs showing promise in ameliorating cardiac reperfusion damages. The work offers a valuable formulation design strategy for polysaccharides. However, from the point of view of the high-throughput (HTP) screening platform, the reviewer cannot evaluate it sufficiently because the information for the HTP platform is lacking. In addition, the overall structure of the HTP system is composed of commercial devices such as pumps, valves, robotic arms, and even the well-known flow-focusing microfluidic chip, so the reviewer cannot find out any novelty and technical advancement in their system. This manuscript cannot give any insight and critical contribution to the HTP microfluidic society.

Answer: Thank you for your time and the efforts in reviewing the current manuscript. First of all, we are grateful for your kind comments to the significance of our work in designing polysaccharides-based formulations for cardiac therapies, and we appreciate your fair and frank comments to the currently adopted high-throughput (HTP) screening platform, and we recognize the importance of providing a comprehensive description of the HTP system to allow you further evaluate its integration and functionality within our research framework.

To address the concerns regarding the detailed information on the HTP platform, we have expanded the manuscript to include a more thorough description of the system's architecture and operational parameters, which are described in the following sections. This augmentation not only details the configuration of commercial components, such as pumps, valves, robotic arms, and the flow-focusing microfluidic chip, but also underscores the innovative integration of these elements to enhance the platform's efficiency and applicability in our specific research context.

While it is true that the individual components of our HTP system may be commercially available, the novelty and technical advancement of our work lie in the customized assembly and optimization of these components.

To the best of our knowledge, our study presents the first time of integrating high-throughput microfluidic synthesis with automated NPs characterization system. This pioneering combination harnesses the precision and scalability of microfluidics alongside the efficiency of automated characterization techniques, offering a comprehensive platform for the rapid and detailed analysis of synthesized particles. This integration marks a significant

advancement in the field, setting a new standard for the synthesis and evaluation of nano/micro-systems in a high-throughput setting.

In line with our objectives, we developed a proprietary control system, custom-designed to fulfill the specified research goals.

This innovative software solution, now patented under patent number: 2023SR0681955, underscores the uniqueness and novelty of our current system. The granting of the patent also partly indicates the potential originality and technological advancement embodied by our integrated approach, further distinguishing our platform in the realm of high-throughput microfluidic systems.

We appreciate your constructive critique and are confident that these revisions have strengthened the presentation and impact of our work.

In terms of the applications of the resultant combinatorial NPs for the targeted cardiac RNAi therapeutics, their research was performed systematically, and some results are interesting. Here are some issues to be addressed to strengthen the manuscript.

Answer: Thank you for your encouraging comments, as well as for the time and effort in helping us improve the manuscript. Specific discussion of your concerns and suggestions are present in point-by-point response below.

1. Even though the authors put some references, it is more helpful to provide additional information about the microfluidics chip, including details such as size, depth, and other relevant parameters (Fig. 2a), and chip fabrication in the Supplementary Materials.

Answer: We sincerely appreciate your constructive feedback and the opportunity to enhance the clarity and completeness of our manuscript regarding the microfluidics chip used in our study.

The initial step involved designing and modeling the chip's structure using SolidWorks (Dassault Systems S.A). The fabrication process employed a projection micro-stereolithography-based 3D printing method provided by BMF Precision Tech Inc., achieving a high resolution of 10 μm . Upon completion of the printing, the chip underwent a rinsing process with isopropyl alcohol (IPA) to eliminate any residual impurities. Subsequently, the chip was cured using a UV curing device to ensure the stability and integrity of the printed structure. Finally, nitrogen gas was used to eliminate any remaining traces of IPA, ensuring the chip was free from contaminants before further application in the synthesis of EEGP-based

NPs. The detailed parameters for the produced microfluidics chip are shown below (**Fig. 1 for Review**).

Fig. 1 for Review. Schematic diagram illustrating the dimensions of the co-flow focusing microfluidic chip. The inner tube outlet possesses an inner diameter of 110 μm , whereas the outer tube has an inner diameter of 1120 μm .

The corresponding information has been included in the revised manuscript as **Fig. S16** and labelled with change track. We hope that these changes satisfactorily address your concerns.

2. In Fig. 2a, the high-throughput screening system is depicted with several inner-phase solutions in different reservoirs. However, it appears that all reactant reservoirs share a common tubing connected to the syringe pump. This design may lead to cross-contamination if some amount of reactant remains in the tubing. Could you provide clarification or discuss any measures taken to address and minimize the potential for cross-contamination in the experimental setup?

Answer: Thank you for bringing up this question and for the time and effort in helping us improve the manuscript.

The inner phase reservoir is designed to accommodate a total of 10 reagent tubes, with five of them containing different inner phase solutions, while the remaining five tubes are allocated for holding the cleaning solution (**Fig. 2 for Review**). The synthesis procedure for the first sample at different flow rates initiates after the initial injection of the inner phase solution into the syringe through the three-way valve. To maintain the integrity and purity of the collected samples, the collection command was executed following the completion of the synthesis process (**Supplementary Video S1**). This enables the injection of residual samples in the pipeline into the well plate, while the synthetic solution at the front end is collected in the effluents collecting box. To avoid cross-contamination between samples associated with different cationic molecules, the cleaning solution from the reagent tube is drawn into the

system, and the entire tube is cleaned after completing the sample synthesis for each reagent tube. A total of 7 distinct flow rate conditions are set for each cation molecule, allowing for the performance of 5 different inner phase solution syntheses per batch. This leads to the acquisition of samples and the corresponding characterization data from 5×7 wells in the 48-well plates. Following that, the solution and well plates of the inner phase reservoir are replaced, and the next batch of synthesis is carried out for the subsequent set of samples.

The corresponding information has been added in the manuscript as **Fig. S2** and **Supplementary Video S1** and labelled with change track. We hope that these changes satisfactorily address your concerns.

Fig. 2 for Review. The representative schematic of the pre-mixed automatic microfluidics system.

3. In the manuscript, a section titled "A robotic-assisted microfluidics platform for nanoparticle formation screening" is highlighted as a key result of the experiment. However, the role of the robotic arm appears to be insufficiently described and emphasized. The authors should provide clarification on the specific functions and contributions of the robotic arm within the context of the manuscript.

Answer: Thank you for your suggestion and comments. In the current study, a robotic-assisted microfluidics platform was applied for nanoparticles formulation screening. The integrated microfluidics platform consists of a sample preparation module, a microfluidic module and an analysis module. The analysis module integrates an automated online characterization system

that combines a robotic arm, two peristaltic pumps, and an optimized dynamic light scattering (DLS) analysis system. The robotic arm ensures precise sampling by maneuvering to various positions on the well plate and collaborates with the peristaltic pump to extract samples from the 48-well plate into the detection cell. Upon completion of the analysis, another peristaltic pump is employed to transfer the solution from the detection cell to a waste liquid cylinder. Moreover, the robotic arm and peristaltic pump can be utilized to collect and dispense deionized water from a diluent storage container. This enables the cleaning of the sample cell or the dilution of the solution under examination. The detailed function of robotic arm is illustrated in **Supplementary Video S1**.

The corresponding information has been added in the manuscript and labelled with change track. We hope that these changes satisfactorily address your concerns.

4. Figure legend for Fig. 3e is missing.

Answer: We really appreciate for your reminder. The legend for Fig. 3e is added to the manuscript, which is labelled with change track in the manuscript.

5. What is the meaning of KD in Fig. 4a?

Answer: Thank you for your query. The abbreviation KD refers to knockdown in **Fig. 4a**. To make it clear, the corresponding information is added to the manuscript, which is labelled with change track. We appreciate your efforts in improving the clarity of the work.

6. The author should furnish more detailed information about the siRNA sequences employed in the experiments.

Answer: Thank you for your kind suggestion. The detailed information of the siRNAs employed in the experiments has been added to the “materials and reagents” section, which is labelled with change track in the manuscript.

RESPONSE TO THE REVIEWER'S COMMENTS

Reviewer #3:

In their paper entitled "Diversity-oriented combinatorial formulation screen for cardiac RNAi therapeutics with polysaccharide framework", Gao et al use a computational approach to generate a series of nanoparticles capable of delivering siRNA to the myocardium following injury. The authors target Dectin-1 for delivery to post-infarcted macrophages and show in vivo knockdown of a pro-inflammatory gene and improvements in cardiac function. The screening process is interesting, yet there are many unresolved questions that dampen enthusiasm.

Answer: Thank you for your time and the efforts from the reviewer in reviewing the current manuscript. The insightful commentaries will undoubtedly help us to improve the overall quality of the manuscript. A detailed response to each of your concerns and suggestions is provided below, addressed on a point-by-point basis.

1. If the authors knew a priori that they wanted to target Dectin-1, why the need for a long screening process rather than working to modify an already existing NP to target this protein?

Answer: Thank you for your constructive feedback and comments and give us chance in further clarifying the principle of design of the current work. The major highlight of the current work is not simply focusing on developing one specific Dectin-1 targeting nanosystem for potential therapeutic applications. Instead, our research is fundamentally aimed at establishing a systematic approach for the identification of polysaccharide-based nanosystems optimized for targeted RNA interference, capitalizing the exclusive recognition of certain glycans by specific lectins to achieve a targeted delivery in a simple engineering method. And the Dectin-1 targeting glycan, β -glucan, is adopted in the current study as a model polymer due to its potential application in targeting lesioned myocardium.

In this context, our study introduces a novel screening protocol characterized by its ability to evaluate diverse chemical structures collectively. This approach, distinguished from traditional methodologies, employing a diversity-oriented combinatorial optimization strategy, enabling the exploration of a wide array of molecular constructs with varied chemical properties. This methodological framework significantly enhances the likelihood of discovering efficacious candidates facilitated the discovery of the GluCARDIA NPs. Remarkably, compared to existing nanosystems with analogous components, GluCARDIA NPs demonstrated gene silencing

capabilities at substantially lower concentrations, necessitating 33% less siRNA and 70% less nanoparticle mass to achieve comparable effects. This efficiency underscores the value of our diversity-oriented screening process, proving its effectiveness in pinpointing optimal formulations for targeted delivery, thereby streamlining the development process for RNAi-based therapies.

An additional benefit of employing a diversity-oriented combinatorial optimization approach lies in covering complex chemical structures, thereby the generated knowledges exhibited higher transferability but not limited to specific type of materials. For example, in interpreting the critical factors governing the biocompatibility of the nanosystem, while previous studies underscored the contributions of hydrophobicity and overall positive molecular charges to biocompatibility assessment of specific materials such as lipids or polymers, our findings propose that in a more general point of view, the balance and distribution of a compound's negative and positive components are equally crucial in influencing its biocompatibility. This perspective offers a refined understanding of the molecular underpinnings features that govern the biocompatibility of cationic compounds, and also reflects the advantage of the currently proposed strategy.

In summary, the current work underpins a broader exploration of chemical space and a deeper understanding of fabricating polysaccharides-based nanosystems potentially for genetic intervention. We believe that this approach significantly contributes to the field, not only by identifying efficient targeting nanosystem but also by offering insights into further nanosystem design.

We appreciate the opportunity to clarify these aspects and hope that our responses address your queries satisfactorily.

2. The *in vitro* work is convincing that nanoparticles are created with siRNA, but testing on just 2 cell types is not convincing. Additionally, if the eventual goal is human translation, the negative effects on THP-1 are not encouraging. Could the authors discuss why they are creating a system that seemingly targets mouse macrophages?

Answer: We thank the reviewer for bringing up this question and for the time and effort in helping us improve the manuscript. For the *in vitro* part, we selected two types of macrophages for gene knockdown evaluation, including RAW 264.7 cells and THP-1 cells, this is mainly based on their distinct Dectin-1 expression profile. For the RAW 264.7 cells, the naïve cell line shows low-to-no expression of Dectin-1 receptor (J Immunol. 2004 Jan 15;172(2):1157-62). Whereas for THP-1 cells, it was indicated that THP-1 derived macrophages showed high

Dectin-1 expression after cytokines stimulation (Eur J Immunol. 2017 May;47(5):848-859). Therefore, in this part, THP-1 cells were selected as model cell line for testing the Dectin-1 targeting efficiency from the developed nanosystem, whereas Dectin-1⁻ RAW264.7 cells were regarded as a side-by-side comparison.

We would also like to clarify that the cellular uptake of GluCARDIA NPs by THP-1 cells were substantially higher than that from RAW264.7 cells (**Fig. 5f**). Correspondingly, the gene silencing efficacy in THP-1 cells were also higher than that from RAW264.7 cells, confirming the Dectin-1 mediated cellular uptake towards GluCARDIA NPs. We hope this explanation could properly address your concern and thank you once again for providing us with the opportunity to enhance the clarity and impact of our work.

3. The in vivo delivery needs much more quantification. A single time-point measured semi-quantitatively is not encouraging. The authors should measure %id/g in each organ as right now the heart seems to have taken up very little of the total NP population.

Answer: Thank you for the valuable suggestion and the time and efforts in helping us improve the manuscript.

First of all, we would like to point out that promoting the myocardium accumulation of NPs is inherently challenging, this is mainly due to the large cardiac output, coupled with the dynamic nature of cardiac contraction–relaxation cycles and the rapid blood exchange during each cycle, limits the exposure time for NPs to interact with the myocardium (*J. Comput. Phys.*, 2016, 305:1065). For example, previous study adopted heart targeting index (HTI, HTI = heart fluorescence emission/liver fluorescence emission) to evaluate the cardiac accumulation of fluorescence labelled NPs, and the results showed the HTI of blank green fluorescence protein (GFP) is less than 0.01, and tannic acid modified GFP NPs could substantially promote the HTI to 0.13, indicative of a satisfactory cardiac accumulation by tannic acid modification (*Nat. Biomed. Eng.*, 2018, 2:304). Notably, despite this enhancement, the cardiac targeting efficiency remained several orders of magnitude lower than hepatic levels, reflecting the inherently preferred clearance of NPs by the liver compared to the heart.

In such context, we checked the accumulation profile of ICG-labelled GluCARDIA NPs in major organs at 8 h post-injection to mice with myocardial IR injury though real-time fluorescence imaging, which is well-established and widely-adopted in previous papers (*Sci. Transl. Med.*, 2020, 12:e1063; *Nat. Mater.*, 2023, 22:391, *Nat. Commun.*, 2014, 5:4880). To quantify the cardiac accumulation of NPs, we similarly adopted the well-established terminology HTI to evaluate the cardiac targeting efficiency from the currently proposed

nanosystem. The HTI value for GluCARDIA NPs was calculated as 0.198 at 8 h, which is comparable to previous studies and indicates a satisfactory cardiac tropism in the setting of myocardial IR injury (*Nat. Biomed. Eng.*, 2018, 2:304; *Theranostics*, 2021, 11:3725). For comparison, we fabricated a dextran-based NPs, which is a type of α -glucan without Dectin-1 binding affinity. Quantitative analysis showed that the HTI value in the ICG-GluCARDIA NPs group was 2-fold higher than that in the ICG-Dextran NPs group, further indicated the distinctive targeting potential of our developed GluCARDIA NPs towards infarct myocardium. Despite the substantial lower cardiac accumulation comparing to hepatic accumulation, we further determined to evaluate its potential impact on gene silencing, and the knockdown efficiency was evaluated by western blotting analysis (**Fig.1 for Review**). Results showed that while the overall accumulation of GluCARDIA NPs in cardiac tissue is significantly lower than that in liver, the gene knockdown efficiency was still comparable to that observed in hepatic tissue. This might be explained by the “siRNA saturation” effect as described by Lieberman *et al.*, as only marginal amount of the siRNA ($\sim 10^{-9}$ pmol) is needed to achieve a maximal gene knockdown efficiency, and further increasing of the siRNA showed no significant difference in terms of gene knockdown yield (*Nat. Biotechnol.*, 2015, 33:870). Therefore, despite the relative cardiac accumulation of GluCARDIA is still substantially lower than that in liver, the currently developed nanosystem can significantly promote the cardiac accumulation and sequentially achieve the envisioned design.

We hope this explanation resolves your concern and reassures the therapeutic potential of the currently proposed nanosystem. The corresponding information has been added in the manuscript as aligned as **Supplementary Fig. 9** and labelled with change track.

Fig.1 for review. Western blotting analysis was performed to evaluate the gene silencing efficacy of GluCARDIA-siGAPDH NPs in heart and liver.

4. There is no timecourse of delivery. The authors delivered 24 hours after IR, but macrophages peak 3-7 days post-injury in most published studies. They do not examine any PK or PD of the NP.

Answer: We thank the reviewer for bringing up this question and for the time and effort in helping us improve the manuscript. In current work, we conducted a therapeutic study involving three *i.v.* injections of 0.1 mg kg⁻¹ siRNA per mouse, the timepoints were set as 30 min, 24 h and 72 h post-reperfusion. The timepoints of three injections were pre-determined according to the expression profile of Dectin-1 post myocardial IR injury, which is the critical homing receptor of GluCARDIA NPs. As reported, Dectin-1 showed time-dependent activation post-IR, initiated at 1 h post reperfusion, peaked at 24 hours and subsided to baseline level at 72 h post-IR (*Circulation*. 2019, 139:663). In addition to the Dectin-1 expression profile, the determination of the therapeutic time window also relies on the expression profile of IRF3 and the maladaptive innate immune response following myocardial IR. Previous studies identified aberrant IRF3 activation in macrophages as a critical regulator of fatal inflammatory responses after ischemic cardiac incidence, and the IRF3 activation was initiated shortly after the ischemic induced cardiac cell death (*Nat. Med.*, 2017, 23:1481). And the IRF3 and the type I IFN regulated deleterious hyper-inflammatory responses will take the course of 3 – 4 days post cardiac injury, sequentially gives a way to reparative phases (*Nat. Rev. Immunol.*, 2018, 18:733). Upon such consideration, the designed therapeutic course in the current study involves three *i.v.* injections of 0.1 mg kg⁻¹ siRNA per mouse, separately at 30 min, 24 h and 72 h post-IR.

Regarding the PK or PD NPs, as we discussed in the manuscript and in the previous section, the principle focus for the current paper is to propose and validate a practical scheme, termed as diversity-oriented combinatorial optimization process, in identifying polysaccharide-based nanosystem for targeted RNAi in a more efficient and controlled manner. This approach was meticulously demonstrated through our self-developed, fully-automated microfluidic platform. In line with this objective, we have further systematically evaluated the *in vivo* gene knockdown efficiency (**Fig. 6i-k**) and therapeutic efficacy (**Fig. 7**) of the identified GluCARDIA system for myocardial IR injury.

Given the comprehensive nature of the current manuscript and its already substantial focus on the development and *in vivo* validation of the GluCARDIA system, further detailed characterization of the PK or PD parameters for GluCARDIA NPs is beyond the scope of the current manuscript. We agree with the reviewer that further exploration into the PK and PD aspects of GluCARDIA NPs would enrich our understanding and support future clinical applicability. However, in order to minimize the time consuming and expensive animal studies, as well as application of the animal 3Rs rules, in an early development stage, we believe the

current studies can already provide necessary information for the assessment of the potential of the GluCARDIA for the proposed biomedical applications.

We hope these clarifications may address your concern and we appreciate your time and effort in providing the comments.

5. There are no in vivo macrophage results or examination of infarct size/cell death. The authors have created a macrophage-targeted material, but do not look at macrophage polarization in vivo or any inflammatory markers. Further, macrophages play a big role in acute cardiomyocyte cell death but this is not examined.

Answer: We thank the reviewer for bringing up this question and for the time and effort in helping us improve the manuscript.

Considering your suggestion, we further performed terminal-deoxynucleotidyl transferase mediated nick end labeling (TUNEL) staining analysis to evaluate the cell death in the lesions of myocardium at 24 h post-administration. As shown in **Fig. 2 for Review**, the reduced TUNEL⁺ cells indicated the cardioprotective effects of GluCARDIA-siIRF3.

Fig. 2 for Review. Representative TUNEL staining of cardiac myocytes in heart sections from IR and GluCARDIA-siIRF3 NPs. Scale bar: 50 μm.

Furthermore, we assessed the transcription levels of key regulatory genes in each group, including TGF-β, interferon-β, and caspase-3. These genes are known indicators in reflecting the fibrosis, immune responses, and cellular apoptosis status following myocardial IR, and their levels were quantitatively analyzed using the qPCR. Comparing to GluCARDIA-siNC or IR group, marked decreases in transcriptional levels of these genes were noted in the GluCARDIA-siIRF3 group, underscoring the potential beneficial effects of GluCARDIA-siIRF3 treatment following myocardial IR (**Fig. 3 for Review**).

We hope the supplementary experiments reassures the therapeutic potency of our developed nanosystem. The corresponding information has been added in the manuscript (Page 25, line 18-27) and labelled with change track.

Fig. 3 for Review. RT-qPCR was used to detect the expression levels of TGF- β , Interferon- β and Caspase-3 in mice from each group. Data are presented as mean \pm SD. ns: not significant; * $p < 0.05$; ** $p < 0.01$; *** $p < 0.001$; **** $p < 0.0001$.

6. If the authors created a better targeting system, they should compare to an existing system as a control. There are countless papers of NP targeting to the heart that the authors could use a control. The premise is that they have done something much better through their approach (which they may have) but do not have any comparisons.

Answer: Thank you for your insightful comments regarding the use of an existing system for comparison on targeted delivery to the injured myocardial.

In light of your suggestion, we compared the subcellular uptake of different NPs after *in vivo* administration. To this end, fluorescein isothiocyanate (FITC) labelled GluCARDIA NPs was synthesized for fluorescent imaging purpose. As a comparison, we also developed FITC-labelled Dextran NPs, as a negative polysaccharide control without Dectin-1 targeting affinity, and FITC-labelled SM-102 LNPs, as a comparative benchmark for commercialized NPs in gene delivery. Mice were subjected to 50 min of ischemia followed by 24 h of reperfusion, afterwards, FITC-labelled GluCARDIA NPs/Dextran NPs/SM-102 LNPs was i.v. injected for further analysis. 8 h post-injection, organs including heart and liver were harvested for flow cytometry analysis. The gating strategy to identify different immune cell subsets by flow cytometry experiments was shown as Fig. 4a for Review.

Comparing to Dextran NPs and SM-102 LNPs, GluCARDIA NPs demonstrated a distinct cellular interaction pattern with cardiac immune cells following myocardial IR. GluCARDIA NPs mainly co-localized with monocytic cells. Meanwhile, we also observed an enhanced monocytic interaction from GluCARDIA NPs comparing to Dextran NPs and SM-102 LNPs. (**Fig. 4b for Review**). Notably, GluCARDIA NPs also exhibited substantially higher recognition by dendritic cells, which are also known as major Dectin-1⁺ cellular subsets upon sterile inflammation (*Nat. Commun.* 2016; 7:12368), while exhibiting a comparable cellular uptake with other immune cells (*Circulation.* 2019; 139:663–678).

Regarding to the potential cellular interaction with hepatic immune cells, despite the major colocalization was still observed in monocytic cells, in contrary to the enhanced monocytic targeting efficiency in lesioned myocardium, GlucCARDIA NPs exhibited a similar monocytic internalization efficiency comparing to SM-102 LNPs (**Fig. 4c for Review**). This may be attributed to the restrained Dectin-1 expression profile in hepatic macrophages upon sterile injury (*J. Control. Release.* 2023; 357:120). In conclusion, here we have demonstrated that in the setting of myocardial IR, our designed GluCARDIA NPs are primarily recognized by monocytic cells, significantly exceeding its uptake in other immune cells. Moreover, GluCARDIA NPs exhibited substantially higher targeting efficiency towards cardiac monocytes comparing to α -polysaccharides (dextran) based NPs and commercialized benchmark NPs (SM-102 LNPs), offering a promise delivery system for efficient cardiac gene delivery post IR.

The corresponding information has been added in the manuscript (Page 22, line 22-34; Page 23, line 1-13) and assigned as **Supplementary Fig. 11** in the **Supporting Information**. All changes are labelled with change track. We hope that these changes satisfactorily address your concerns.

Fig. 4 for Review. (a) Gating strategy for flow cytometry analysis. All cells were extracted from the heart and liver, and further labeled with antibodies as described in Methods. Monocytes was identified as CD45+CD11b+Ly6G-Ly6clow to high, neutrophils were identified as CD45+CD11b+Ly6G+, T cells were identified as CD45+CD11b-CD3+, B cells were identified as CD45+CD11b-B220+, NK cells were identified as CD45+CD11b-NK1.1+, DCs were identified as CD45+CD11b-CD11c+MHC II+. Quantitative analysis of immune cell uptake of SM-102 LNPs/Dextran NPs/GluCARDIA NPs was performed on (b) heart and (c) liver. Data are presented as mean \pm SD. ns: no significance; *, $p < 0.05$, **, $p < 0.01$, ****, $p < 0.0001$.

REVIEWERS' COMMENTS

Reviewer #1 (Remarks to the Author):

The authors have addressed my concerns, and I now recommend publication. Congratulations!

Reviewer #3 (Remarks to the Author):

The authors have addressed many of the concerns. While it would have been great to add time course data since macrophages peak later than delivery, the reviewer understands the challenges with adding more animals.

Reviewer #4 (Remarks to the Author):

The author has made significant efforts to address the comments provided by Reviewer 2, which is commendable. However, upon reviewing the author's responses, there are still several comments that require further attention. By addressing these remaining comments, the author can enhance the overall clarity, depth, and quality of the manuscript. Therefore, the author is encouraged to revisit these comments.

1. What could be the real-time implications of formulated nanoparticle performance if the GluCARDIA NPs are used in cardiac RNAi therapy? The author put this as their aim in both the abstract and introduction. Hence, it should be tested.
2. How is real-time monitoring of the distribution and behavior of nanoparticles in the body being conducted?
3. What imaging techniques can be used other than fluorescence to track nanoparticles' movement and accumulation in target tissues? What are the potential applications of using fluorescent imaging?
4. What are the limitations of the developed technique for NP formulations over the existing techniques?

5. What were the key findings regarding the effectiveness and safety of GluCARDIA NPs in delivering therapeutic RNA for cardiac reperfusion damage?
6. How generalizable is the formulation design strategy for other therapies beyond cardiac RNAi therapy?
7. The author should provide more specific data or metrics on the improvements in long-term prognosis achieved by using GluCARDIA NPs.
8. What are the potential limitations of the used approach, and how could future research build upon your findings?

RESPONSE TO THE REVIEWER'S COMMENTS

Reviewer #4:

The author has made significant efforts to address the comments provided by Reviewer 2, which is commendable. However, upon reviewing the author's responses, there are still several comments that require further attention. By addressing these remaining comments, the author can enhance the overall clarity, depth, and quality of the manuscript. Therefore, the author is encouraged to revisit these comments.

Answer: Thank you for your time and the efforts in reviewing the current manuscript. We are grateful for your kind feedbacks and the insightful commentaries will undoubtedly help us to improve the overall quality of the manuscript. Specific discussions of your suggestions are present in point-by-point response below.

1. What could be the real-time implications of formulated nanoparticle performance if the GluCARDIA NPs are used in cardiac RNAi therapy? The author put this as their aim in both the abstract and introduction. Hence, it should be tested.

Answer: Thank you for bringing up this question and for the time and effort in helping us improve the manuscript.

We would like to clarify that our study did not specifically investigate the real-time implications of the developed GluCARDIA NPs in the context of myocardial ischemia-reperfusion (IR). Instead, we focused on its implication at specific time-points in several distinct aspects: 1) We examined the cellular interaction patterns of different nanosystems with cardiac immune cells following 24 h of reperfusion; 2) We assessed the biodistribution of GluCARDIA NPs at a 24 hours post-reperfusion; 3) We evaluated the gene knockdown efficiency by measuring the protein expression levels of IRF3 and phosphorylated IRF-3 (p-IRF3) after therapeutic intervention at day 4 post-reperfusion.

As such, in the abstract and introduction sections, we avoided using terms "real-time analysis" to accurately reflect the scope of our investigations.

We acknowledge the importance of clarity in scientific writing. Therefore, upon your prompt, we have carefully reviewed other sections of the manuscript to ensure consistency and coherence in our descriptions. We are grateful for your constructive feedback, which has provided us with the opportunity to enhance the clarity and precision of our work.

2. How is real-time monitoring of the distribution and behavior of nanoparticles in the body being conducted?

Answer: Thank you for giving us chance in clarifying the experimental detail of the current work. In this study, we did not monitor the real-time bio-distribution of the GluCARDIA NPs in vivo. Instead, as we described in the manuscript, the biodistribution of GluCARDIA NPs was tested at specific time-point, by i.v. injecting indocyanine green labelled GluCARDIA NPs (ICG-GluCARDIA NPs) at 24 h post the reperfusion. After 8 h, major organs including heart, lung, liver, spleen and kidney were harvested for fluorescence imaging. The cardiac accumulation of GluCARDIA NPs was calculated based on a previously established term (Nat. Biomedical. Eng., 2018, 2, 304), heart-targeting index (HTI, HTI = heart fluorescence emission/liver fluorescence emission). We hope this explanation could properly address your questions.

3. What imaging techniques can be used other than fluorescence to track nanoparticles' movement and accumulation in target tissues? What are the potential applications of using fluorescent imaging?

Answer: Thank you for bringing up this question. There are different imaging modalities applied in exploring the biodistribution of NPs and the cell–NPs interactions in vivo other than fluorescence imaging, including magnetic resonance imaging (MRI), positron emission tomography (PET), single photon emission computed tomography (SPECT), and computed tomography (CT) with suitable design and construction of contrasting reagents.

In general, MRI imaging exhibited the most optimal tissue penetration, satisfied biosafety, sensitivity (requiring the contrasting reagents with the concentration 10^{-3} - 10^{-5} mol/L) and tissue resolution (10 – 100 μ m). For example, Senders *et al.* fabricated high-density lipoprotein (HDL) NPs with a lipophilic perfluoro-crown ether (PFCE; ^{19}F -HDL) payload to allow investigating the interaction between myeloid cell and NPs via hot-spot ^{19}F MRI imaging (Nat. Nanotechnol. 2020; 15, 398). Due to advantages of MRI imaging in high resolution, it can also allow imaging in single cell level (Magn. Reson. Med. 2006; 56, 1001). However, it also exhibited limitations such as exorbitant price, complex procedure and extended measurement time (it usually takes hours for imaging one mice). Moreover, due to its long imaging duration, it also lacks sufficient temporal resolution to track the moving entity in vivo.

Nuclear imaging, including PET-CT and SPECT, is achieved by modifying NPs with radiotracers (e.g. ^{89}Zr , ^{68}Ga) for tracking the in vivo biodistribution of NPs. PET imaging exhibited higher tracer sensitivity comparing to MRI, with only requiring the tracer concentration in the level of picomolar (BMB Rep. 2022; 6, 267). As a typical example, Nahrendorf *et al.* adopted ^{64}Cu modified iron NPs to visualize its interaction with macrophages (Circulation 2008; 117, 379). Considering the temporal resolution of normal PET can be

reduced to several minutes, trajectory reconstruction algorithm can be further adopted for tracking the movement of the imaging entity (IEEE Trans. Med. Imaging. 2015; 34, 994). However, nuclear imaging also has limitations including high costs and complex procedures in labelling the NPs with radiotracer. Meanwhile, the existence of radioactive material will inevitably generate biosafety concerns during the handling process.

Among the various imaging modalities, optical imaging techniques, such as fluorescence imaging or bioluminescence imaging, are widely utilized in preclinical settings due to its simplicity, low cost, safety, and convenience compared to other methods. However, fluorescence imaging faces significant limitations for *in vivo* applications, including issues with autofluorescence and limited imaging depth (Nat. Methods, 2009; 6, 465–469). Specifically, in cardiac imaging, the anatomical location of the heart and the relatively low cardiac accumulation of NPs pose challenges for direct, non-invasive, real-time imaging using infra/near-infrared fluorescent dyes (*e.g.*, ICG, Cy7). Therefore, previous studies typically assess the cardiac accumulation of NPs at specific time points through an *ex vivo* manner on harvested organs (Nat. Nanotechnol. 2020; 15, 398). Nonetheless, there are other alternative methods in exploring the fluorescence imaging for real-time monitoring of NPs *in vivo*. Typical example includes using advanced intravital microscopy (IVM), which allows directly visualize cell–NPs interactions using single and multiphoton fluorescence imaging (Nat. Rev. Cardio., 2021; 18, 617). For example, Fish *et al.* adopted IVM to achieve real-time observing NPs mobility and adhesion on inflamed endothelial walls *in vivo* in mesentery veins in mice (Sci. Adv., 2021; 7, eabe0143). Taking advantage of the high resolution of the microscopy, the authors can precisely quantify the amount of absorbed NPs per mm² of vessel walls.

In conclusion, based on the scientific requirements, experimental design, physiochemical characters of NPs and most important, funding and instrument availability, different *in vivo* imaging modalities can be adopted for tracking the biodistribution of administrated NPs.

4. What are the limitations of the developed technique for NP formulations over the existing techniques?

Answer: Thank you for your constructive comment regarding the limitations of our developed technique for NPs formulations compared to existing techniques.

While our diversity-oriented combinatory formulation screening scheme offers significant advantages, such as the ability to identify potent gene delivery cargos through the integration of high-throughput microfluidics and computational analysis, several limitations should also be acknowledged:

1). From the formulation screening point of view, in our current research, we mainly adopted a microfluidics-based high-throughput platform due to its seamless integration between formulation screening and production, industrial transferability, scalability, and low batch-to-batch variation. However, in terms of high-throughput efficiency, this platform is less efficient comparing to a previously reported liquid-handling deck-facilitated straightforward nanoprecipitation method, which have been reported to simultaneously produce 384 types of formulations (Nat. Nanotechnol. 2021; 16, 725). Although this nanoprecipitation-based screening process is mainly suitable for small-volume particle production, it demonstrates clear advantages in screening efficiency;

2). From the computational aided formulation design point of view, as we also acknowledged in the manuscript, the limited dataset size poses challenges in designing a robust predictive algorithm for cationic compounds selection or design using conventional cheminformatics methods. Our objective was rather to extract scientific insights from the screening process and provide perspectives for interpretation methods. Unlike studies that employ complex algorithm including machine learning or neural networks to predict formulations with optimal characteristics (PNAS, 2019; 116, 11259), we refrained from using sophisticated machine learning algorithms due to the high risk of overfitting with the current size of dataset. Additionally, the high number of generated molecular descriptors relative to the small database size could also lead to the “curse of dimensionality” effect. Consequently, our current model may fall short in predicting superior formulations but rather to understand the potential factors influencing the observed phenomena.

Notwithstanding the limitations, we believe that our technique represents a significant step forward in the field of precision cardiac therapy, offering a robust platform for the systematic exploration and optimization of gene delivery systems.

5. What were the key findings regarding the effectiveness and safety of GluCARDIA NPs in delivering therapeutic RNA for cardiac reperfusion damage?

Answer: Thank you for bringing up this question and we appreciate the opportunity to clarify the some of the key findings in the current study.

The main focus of the current work lies in employing a fully automated microfluidic platform to screen and validate effective gene delivery vehicles through a non-biased approach. This methodology enabled us to systematically evaluate over 161 formulations, leading to the identification of an optimal formulation with corresponding screening process elaboration.

With the similar building blocks and fabrication method, the newly identified GluCARDIA NPs, a result of this expansive screening, demonstrated clear advantages comparing to conventional formulation optimization process by achieving superior gene silencing efficacy with a 33% reduction in siRNA dosage and 70% less NPs mass (Journal of Controlled Release 357 (2023) 120–132). This underscores the efficacy of a diversity-oriented screening approach in identifying optimal conditions in a more efficient and robust manner. Our findings contribute a practical framework for the development of polysaccharide-based nanosystems for targeted RNA interference. The methodology and the resultant formulation not only enhance the efficiency and control of the delivery process but also pave the way for the development of novel active targeting NPs focused on myocardial repair and protection. We believe these aspects significantly underscore the novel contributions of our current study, distinct from our previous works, and provide meaningful advancements in the field of nanoparticle-based gene delivery.

We hope this response clarifies the novelty and significance of our research.

6. How generalizable is the formulation design strategy for other therapies beyond cardiac RNAi therapy?

Answer: We thank the reviewer for bringing up this question. The term “polysaccharide-framework” in our research refers to a simplified strategy of adopting specific polysaccharide as major building block to fabricate active targeting nanosystem without further modification. This is mainly achieved by leveraging the exclusive recognition of certain glycans by specific lectins, which may be abnormally expressed in lesioned sites, enabling a targeted delivery without subsequent modifications. For example, in our current study, we adopted β -glucan as fixed building block to construct cardiac targeting delivery, capitalizing the aberrant emergence of Dectin-1⁺ macrophages in injured myocardium upon cardiac IR (Circulation. 2019; 139, 663). Considering the emergence or aberrant expression of Dectin-1⁺ macrophages can also be observed in other cardiovascular diseases such as atherosclerosis (Atherosclerosis, 2015; 239, 318), the adoption of the current strategy may also be transferable towards the atherosclerosis conditioning. As a proof of concept, in our recent work, we also observed a clear accumulation of β -glucan based NPs in atherosclerotic region, partly confirming the transferability of the current strategy towards other diseases model beyond cardiac delivery (**Figure R1, data not published**).

Figure R1. Transferability of the current strategy to other disease scenario. **a)** Expression of Dectin-1 in atherosclerotic region. Scale bar equals 50 µm; **b)** β-glucan based NPs can target and accumulate in atherosclerotic region, as indicated by the arrow. **All data are not published.**

Apart from β-glucan, the current strategy can also be presumably transferred to other types of glycans based on their lectin selection and disease model. For example, sulphated polysaccharide, such as dextran sulphate or fucoidan, can be specifically recognized by P-selectin to target platelets (Biochim Biophys Acta, 2009; 2, 141), and the polysaccharides sulphate based NPs can also be feasibly constructed through inotropic gelation method (Colloids and Surfaces B: Biointerfaces, 2005; 44, 143), indicating the potential transferability of the current method to target platelets-rich region, including tumor.

In conclusion, the formulation design strategy we have developed is inherently generalizable and holds significant potential for advancing a wide range of therapeutic modalities beyond cardiac RNAi therapy. Further studies are underway to explore and validate its application in other areas, reinforcing its versatility and robustness. Nonetheless, as we also acknowledged in our manuscript, further in-depth exploration of a wider array of polysaccharides-based formulations is essential to develop a precise predictive algorithm that can facilitate the creation of innovative nanosystems.

7. The author should provide more specific data or metrics on the improvements in long-term prognosis achieved by using GluCARDIA NPs.

Answer: Thank you for your constructive comment for providing specific data on the long-term prognosis improvements achieved with GluCARDIA NPs. In our original manuscript, we conducted several key experiments to evaluate the long-term effects of our therapy.

1). Cardiac function: At 4 weeks post-reperfusion, echocardiograph (ECG) analysis was conducted on different groups for evaluating the cardiac function. As can be seen from **Fig. 7e-f** in the manuscript, the ejection fraction (EF%) from GluCARDIA-siIRF3 group is substantially higher than that from IR group ($p < 0.001$). Meanwhile, GluCARDIA-siIRF3 also

ameliorated the left ventricular (LV) remodeling as reflected by the significantly reduced left ventricular internal diameter end systole (LVIDs, $p = 0.039$), indicating an ameliorated prognosis after 4 weeks.

2). LV remodeling. Histological assessments were also performed 4 weeks post-reperfusion to reflect the LV remodeling upon long term prognosis. Masson's trichrome staining of heart section from each group was also conducted at day 28 post-IR (**Fig.7g in manuscript**), demonstrating a preserved wall thickness (**Fig. 7h in manuscript**) and lower percentage of fibrosis in the left ventricle (**Fig. 7i in manuscript**) in mice treated with GluCARDIA-siIRF3. Adverse remodeling post myocardial IR injury can lead to cardiomyocytes hypertrophy, we observed that mice treated with GluCARDIA-siIRF3 showed considerably decreased myocyte cross-sectional area in comparison to GluCARDIA-siNC or IR (**Fig. 7j in manuscript**).

These findings collectively demonstrate that the use of GluCARDIA NPs leads to significant improvements in long-term cardiac remodeling and function, underscoring their potential for enhancing long-term prognosis in cardiac therapy. We also acknowledge that further evaluation of long-term prognosis over an extended period would provide valuable insights into the therapeutic benefits of the system. However, given that the current study has already partly validated the prognosis of GluCARDIA-siIRF3 treatment at 4 weeks, which is a typical time point for assessing mid-to-long term outcomes, and considering the current workload, this extended evaluation is beyond the scope of the current study.

8. What are the potential limitations of the used approach, and how could future research build upon your findings?

Answer: Thank you for your constructive suggestions. We acknowledge that while our approach has demonstrated some preliminary promising results at the current stage, there are also limitations and areas for improvement that future research can address.

1). Degree of high throughput. Despite in our current study, we achieved the automatic screening of over 160 types of formulations, in our discussion part in the manuscript, we also acknowledged that the current size of dataset was still relatively limited, therefore, the establishment of a robust algorithm for cationic compounds prediction by complex algorithm (such as machine learning) is challenging and scientifically inadequate. Upon such consideration, further efforts should be made in further improving the high throughput degree

of the current platform in accelerating the formulation screening efficiency, which may lay a robust foundation for further formulation identification and prediction.

2). Integration with biological models. Although our method provides valuable insights into formulation optimization, the current method still lacks an automatic manner in screening the knockdown efficiency, which is the most important parameter for efficient gene delivery system design, of produced nanoformulations. In the current study, the test and validation of the critical biological parameters, including biocompatibility and gene knockdown efficiency were still accomplished by manual work. Therefore, further integration of microarrays for high-throughput evaluation of gene knockdown efficiency of produced nanoformulations are desired, yet unfulfilled in the current system.

3). Scalability. While our platform excels in automatic formulation screening and small-scale formulation, scaling up the identified optimal formulations for industrial level production, and the corresponding maximum production capacity were not tested in the current study.

4). Technical expertise. Despite from our current study, we proposed molecular dynamics simulations may be adopted as a valuable supplementary tool for initial evaluations and insights into the inotropic gelation based NPs formation process and siRNA encapsulation efficiency, which is mainly reflected by separately observing the Coulomb's or LJ potential during the simulation. Yet, this demands a certain level of expertise with computational analysis, which might not be readily available in all research settings. Therefore, a simplified algorithm with wider availability towards a broad spectrum of readers should be further proposed. One typical example comes from the paper *Nat. Mater.*, 2018; 17, 361., where the authors generalized the highly specialized molecular descriptor SpMAX4_Bh(s) into a self-coined molecular descriptor NHISS, which can be feasibly calculated by simply counting the number of specific groups, such as fluorine or nitro, in the molecules. However, the establishment of such model also require the increased size of data-set, which should be solved by further increase the degree of high throughput of the platform.